# Novel Biotherapeutics Targeting Biomolecular and Cellular Approaches in Diabetic Wound Healing

**DOI:** 10.3390/biomedicines11020613

**Published:** 2023-02-18

**Authors:** Suraj Kumar Singh, Shradha Devi Dwivedi, Krishna Yadav, Kamal Shah, Nagendra Singh Chauhan, Madhulika Pradhan, Manju Rawat Singh, Deependra Singh

**Affiliations:** 1University Institute of Pharmacy, Pt. Ravishankar Shukla University, Raipur 492010, Chhattisgarh, India; 2Raipur Institute of Pharmaceutical Educations and Research, Sarona, Raipur 492010, Chhattisgarh, India; 3Institute of Pharmaceutical Research, GLA University, Mathura 281406, Uttar Pradesh, India; 4Drugs Testing Laboratory Avam Anusandhan Kendra, Raipur 490024, Chhattisgarh, India; 5Gracious College of Pharmacy Abhanpur Raipur, Village-Belbhata, Taluka, Abhanpur 493661, Chhattisgarh, India

**Keywords:** wound healing, molecular events, mRNA, therapeutic targets

## Abstract

Wound healing responses play a major role in chronic inflammation, which affects millions of people around the world. One of the daunting tasks of creating a wound-healing drug is finding equilibrium in the inflammatory cascade. In this study, the molecular and cellular mechanisms to regulate wound healing are explained, and recent research is addressed that demonstrates the molecular and cellular events during diabetic wound healing. Moreover, a range of factors or agents that facilitate wound healing have also been investigated as possible targets for successful treatment. It also summarises the various advances in research findings that have revealed promising molecular targets in the fields of therapy and diagnosis of cellular physiology and pathology of wound healing, such as neuropeptides, substance P, T cell immune response cDNA 7, miRNA, and treprostinil growth factors such as fibroblast growth factor, including thymosin beta 4, and immunomodulators as major therapeutic targets.

## 1. Introduction

A wound is a loss of continuity in tissues due to injury or abrasion. A wound can result from direct tissue incision through a surgical knife or from broad tissue damage (such as trauma and burns). A wound arises from a bruise, hematoma, abrasion, or laceration. Wounds also result in a fracture of epithelial tissue that alters the function and structure of the skin. The integrity of the skin must be restored, as it has a vital role in the management of homeostasis. Disruption of the integrity of the skin, mucosal layer, or organ tissue causes a domino effect in wound development [1]. Considering the complicated nature of the healing cascade, it is incredible how often it functions devoid of any problems. This mechanism can be hampered by many factors, resulting in delayed wound healing. In acute wound healing, a well-planned sequence of events was followed. A healed wound is characterised by complete connective tissue repair and regeneration into normal function and anatomical structure.All these activities have occurred in a cascade that is associated with the different phases of wound healing [2]. In response to the tissue injuries, four main overlapping stages have been induced, which are hemostasis, the inflammatory stage, the proliferative phase, remodelling, and scar maturation. As many studies have already detailed the mechanism of wound healing in a proper descriptive way, in this review, we have focused on distinct molecular and cellular events and their applicability as impending treatment targets in the management of diabetic wound healing.

### 1.1. Phases of Wound Healing

A sequence of events regulating wound healing mechanisms is depicted in Figure 1. It started when platelets and cytokines caused vasoconstriction and formed a hematoma, reducing the loss of blood in the affected region. The proximity of the wound ends encourages the development of clots and prevents infection by forming a scab. Furthermore, the wounds were filled with fibrin.

A cellular inflammatory reaction serves to destroy all pathogens and dead cells present. Cytokines produced by inflammatory cells allow the fibroblasts to proliferate and the tissue to granulate. Growth mediators stimulate angiogenesis, causing the granulation tissue to develop further; the collagen formation by fibroblasts causes the wound to be healed after about a week, and at the bottom of the wound there is the formation of granulation tissue. This is an essential step, as the epithelial tissue can only propagate and regenerate once. Granulation tissues cover the wound to the extent of the original epithelium; at this level, the wound is completely covered by epithelia. Devascularization of the inflammatory reaction starts to repair itself, wound reduction occurs in the area, and apoptosis of fibroblasts occurs [3]. The various functions of cells and growth factors in wound healing are summed up in Table 1.

### 1.2. Complications in Diabetic Wounds

Diabetic wounds are a major cause of leg ulcers and diabetic ulcers. Diabetes is responsible for the deceleration of the wound-healing process, i.e., hemostasis, remodelling, replication, and inflammation. It influences the day-to-day life, morbidity, and impermanence of the patients. Diabetic wounds due to a deferred, partial, or uncoordinated healing route are described as delayed acute and chronic wounds revealing compromised creativeness. Diabetic wounds exhibit a recurrent inflammatory process, an impediment to mature granulation tissue development, and a decline in wound tensile strength, which is attributed to ischemia resulting in vascular disruption [4]. Each wound is a health alert and demands instant treatment. Generally, based on their genesis, wounds are of two kinds: external and internal. Cuts, fractures, burns, and bruises are examples of external wounds. Owing to peripheral neuropathy, the external abrasions remain unnoticeable to diabetic patients. Internal wounds such as ulcers and calluses have a high risk of bacterial infection and destroy nearby tissues and skin. Hyperglycemia, chronic inflammation, micro- and macrocirculatory dysfunction, hypoxia, autonomic and sensory neuropathy, and altered neuropeptide signalling all affect the wound-healing process in diabetic patients.

#### 1.2.1. Impaired Neuropeptide Wound Healing

Following the initial damage, peripheral nerve fibres of the skin are activated, which causes them to release several neuropeptides into the wound’s milieu. Neuropeptides such as substance P, neuropeptide Y (NPY), and calcitonin-gene-related peptide (CGRP) have an impact on mast cells, endothelial cells, fibroblasts, and keratinocytes, and are implicated in vasoregulation and angiogenesis. Diabetes has been reported to lower the expression of several neuropeptides. Additionally, the poor healing seen in diabetic wounds could be caused by their altered expression and function. The Lewis triple-flare response, which involves nerve-axon-related vasodilation, is diminished or nonexistent in diabetic neuropathy. To release vasodilators such assubstance P, NPY, CGRP, catecholamines, and histamine that cause vasodilatation and hyperaemia during the stimulation of c-nociceptive fibres, neighbouring fibres should be retrogradely stimulated [5].

#### 1.2.2. Vascular Dysfunction

Peripheral artery disease reduces blood flow to the lower limbs and the microcirculation, which both influence wound healing. Additionally, it has been discovered that abnormalities in the microcirculation are related to diabetic neuropathy. In comparison to diabetes patients without peripheral neuropathy, the oxygen saturation, which is determined by oxygen delivery and oxygen extraction, is lower in diabetic patients with peripheral neuropathy. Even though cholinergic iontophoresis induces endothelium-dependent vasodilation, this drop in oxygen saturation still occurs. Furthermore, regardless of whether or not macrovascular disease exists, diabetic neuropathy is associated with decreased endothelial-dependent and endothelial-independent vasodilation and, as a result, compromised microcirculation [6].

#### 1.2.3. Immune System

Altered phagocytic activity and leukocyte malfunction in diabetes individuals have been linked to impaired immune cell function. Because they create and release cytokines, are affected by the local microbiota, and coordinate the shift from the inflammatory to the proliferative phase, macrophages are extensively studied when researching the immune system in chronic wounds in both humans and animals with diabetes. M1 macrophages in acute wounds are replaced by M2 once the inflammatory stage is resolved, whereas M1 macrophages remain to dominate the wound microenvironment in DFUs [7]. Similar to this, prolonged inflammation in diabetes patients results in T-cell accumulation, which may be the cause of elevated levels of TNF- and C-C motif chemokine receptor 4 (CCR4) chemokines, which have a substantial impact on the immune response and facilitate disease progression [8]. Additionally, research has revealed that the immune response deficit in DFUs, as evidenced by the dysregulation of IL-6, MIF, IP-10, and the interferon-inducible protein (MIP)-10, as well as the reduced neutrophil response [9], may contribute to the severity of DFUs. High platelet-to-lymphocyte (PLR) and neutrophil-to-lymphocyte (NLR) ratios have recently been proposed as potential biomarkers for the severity of diabetic foot ulcers [10]. High PLR levels reflect increased platelet activity, inflammation, and the risk for thrombosis and atherogenesis, whereashigh NLR levels result in the upregulation of cytokines and proteolytic enzymes that can result in tissue damage.

### 1.3. Approaches towards Dealing with Complications in Diabetic Wounds

In the last 30 years, molecular biology and various new biotechnological approaches have been developed to assist us in distinguishing between wound healing on damaged human skin and those disturbed by external elements such as physical, synthetic, heat, natural, and diseased conditions such as diabetes. The investigation of novel pathways and molecular possibilities will open up an emerging window into the understanding and therapeutic targeting of the wound-healing process. A few pathways synergistically guiding cell energy, enzyme potential, and biology are related to the remediation of intensive and uninterrupted injuries from skin obstruction. Such pathways give an epigenetic and inherited output that controls neural wound healing. Growth factors (such as VEGF, FGF, nerve and hepatocyte growth factor, and connective tissue growth factor), immunomodulators, neuropeptides, thymosin-4, lactoferrin, T cells, complementary DNA 7, neuropeptide Y, substance P, treprostinil, and homeobox genes are among the molecular targets involved in wound healing. During wound healing, epigenetic regulations regulate the functions of fibroblasts and keratinocytes. In this review, we provide a brief introduction to the epigenetic mechanism of regulating tissue repair in diabetic wound healing. Recent research has revealed that microRNA (miRNA) is an essential regulator of cellular pathology and physiology, making them a viable diagnostic and therapeutic tool. Consequently, the new expertise for wound healing has been grounded on the development and delivery of miRNAs, which indeed will help in wound healing.

## 2. Diabetic Foot Ulceration and Delayed Wound Healing/Mode of Impaired Diabetic Wound Healing

The non-healing diabetic foot ulceration is caused by the effects of multiple genes and/or by their interactions with the environment under the convergence of various abnormalities, including coronary neuropathy, peripheral artery disorder, and an auto-immune disorder. The immune system of the diabetic patient is highly compromised due to a lesser activation of the immune system in comparison with non-diabetic patients. As a result, the number of immune fighter cells recruited in the wounded site is reduced, which often slows the healing process. Along with these factors, diabetic patients have a reduced ability to fight off invading bacteria which enhance the rate of infection at the wound site with delays in treatment. A diabetic person’s foot skin can rupture at any time, putting them at risk of having foot amputation that frequently occurs after neuropathy. When combined with a weakened ability to fight infection, these people are primarily unable to produce a sufficient inflammatory response. As a result, diabetic foot ulcer, which may appear to be a healing wound, serves as a gateway for an infection that might cause sepsis and require limb amputation [10]. For diabetic patients, the above factors function together to induce ulceration and infection. Neuropathic edema leaves the diabetic foot prone to infection and ulceration, and blood flow is decreased by ischemia caused by occlusive arterial disease, both of which lead to delayed diabetic wound healing. In addition, in diabetes, the mobilisation of endothelial precursor cells is disrupted due to a decrease in nitric oxide development that ultimately results in compromised angiogenesis [11]. However, the most significant effect on wound healing is the altered response of immune cells, including neutrophils, platelets, macrophages, endothelial cells, keratinocytes, and fibroblasts, that contribute to the failure of the normal abrasion healing progress [12].

### 2.1. Platelets

A platelet plug is originally formed by platelets, which are important components of hemostasis and are conscious of some of the initial steps in the body’s response to damage. However, for re-epithelialization to take place as the healing process advances, the clot must be destroyed. Additionally, it has been found that under diabetes conditions, platelets are less responsive to the NO generated by the vascular endothelium, which typically decreases aggregation at the artery wall [13]. Imperfection in insulin signaling makes the vascular endothelium less capable of producing NO, which worsens the situation. Thus, diabetes-related platelet dysfunction results in the development of established microvascular disease in diabetic individuals [14]. As a result, throughout the healing process, the platelet phenotype brought on by diabetes impairs wound healing by preventing hemostasis coagulation.

### 2.2. Neutrophils

Neutrophils are essential components that act to kill pathogens throughout the inflammatory phase of wound healing. Increased neutrophil protease expression is reported to involve chronic inflammation and slow wound healing. As well as the proteases, neutrophils liberate extracellular traps (NETs) composed of decondensed chromatin lined with cytotoxic proteins for the termination of microbes. A protein known as peptidyl deaminase 4 (PAD4), responsible for the formation of the NET, is reported to increase the number of neutrophils in humans and diabetes-induced mice, making them more resistant to NETosis and potentially leading to importunate inflammation and tissue damage in diabetes [15].

### 2.3. Monocytes/Macrophages

Monocytes are recruited very early to the abrasion site during the inflammatory process. Monocyte/macrophage infiltration is important for the establishment of the initial inflammatory process as well as for the promotion of the transformation from a pro-inflammatory into an anti-inflammatory setting. During diabetic wounds, this transformation does not happen, and macrophages stay in a continual inflammatory state where they promote overt and indirect destruction of the underlying tissue by attracting other pro-inflammatory immune cells [16]. Recently, it has been found that Ly6CHi monocytes and macrophages increase inflammation. In diabetic wounds, the second form of Ly6CHi macrophages is recruited during the remodeling phase, and there is no shift to anti-inflammatory Ly6CLo cells that lead to unrelenting pro-inflammatory conditions. Moreover, it has been observed that hyperglycemia and the progression of AGEs impair macrophages’ phagocytic potential to clear apoptotic neutrophils, thus encouraging a prolonged pro-inflammatory response. Neutrophil clearance is partly triggered by the conversion of a pro-inflammatory state to an anti-inflammatory state, which enhances pro-inflammatory macrophages at the wounded site [17]. Additionally, research has demonstrated that modifications in the expression of P-selectin and macrophage chemically attractive protein 1 (MCP-1) are responsible for the persistence of monocyte and macrophage access in diabetic animal models [18].

### 2.4. Endothelial Cells

Endothelial cells are positioned around the luminous shell of the blood vessels to handle vasoconstriction and dilation by vasoactive factors such as eNOS. Lower eNOS idiom is linked to the peripheral artery, and peripheral neuropathy disease contributes to the reduction inblood supply from the periphery resulting from poor wound healing. In addition, EPCs (endothelial progenitor cells) are necessary for angiogenesis in wounds, and eNOS promotes EPC migration from the bone marrow, thus leading to the detected decline in eNOS, which contributes to local neovascularization impairment caused bydiabetes [19].

### 2.5. Fibroblasts and Keratinocytes

Fibroblasts and keratinocytes are required to accomplish wound healing. Several variations in the activity of keratinocytes lead to decreased epithelialization of non-healing diabetic wounds, including impaired proliferation and migration of keratinocytes, anomalies of gap junctions, chronic inflammation and infection, decreased angiogenesis, erratic MMPs expression, and oxidative stress. Changes in fibroblast activity aided faulty epithelialization as well as the slow healing of diabetic wounds. Lessened proliferation, accelerated apoptosis, and weakened migration to the wound site are often the most relevant of these. Transformations of all keratinocytes and fibroblasts may indeed be caused by hyperglycemia and AGE creation [20].

## 3. Epigenetics of Wound Healing and Its Response

Epigenetics is characterised as innate transcriptional variations that are not instigated by genetic code modifications. There are three primary methods of epigenetic gene regulation: DNA modification, biochemical histone tail modification, and ATP-based chromatin transformation. These mechanisms are closely interdependent and operate together to either interrupt or allow particular genes to be transcribed. Genes are normally silenced during normal progression, and activation occurs in a specific manner within tissues or cells. However, in certain diseased situations, the epigenetic mechanism can also become deregulated, with severe consequences and complications [21]. Downstream expression of the immunological mediator in monocyte-derived macrophages and immune cells has been controlled by epigenetic alterations in wound healing. Numerous investigations have suggested that macrophage polarisation is influenced by site-explicit histone methylation [22]. The macrophage-specific combined lineage leukemia 1 (Mll1) mutation is associated with a decrease in pro-inflammatory gene expression, whereas the absence of the minute and homeotic disc protein-1-like (Ash1l) gene promotes an anti-inflammatory phenotype by inhibiting the development of IL6 and TNF-α. In pro- and anti-inflammatory macrophage phenotyping activation, H3K27 demethylase, Jumonji domain-restrain protein 3 (JMJD3), plays a significant role. In murine macrophages, LPS and IL-4 mutually up-regulate Jmjd3, along with the manifestation of pro-inflammatory genes and IL-4 target genes, correspondingly. JMJD3 inhibits the formation of pro-inflammatory cytokines induced by LPS in primary human macrophages, and another H3K27 demethylase, UTX, is reduced while orienting IRF4-induced anti-inflammatory differentiation in Jmjd3 knockout mice. Moreover, JMJD3 controls macrophage polarisation in a particular manner depending on the specific tissue. It has been found that neutrophil activity is epigenetically regulated. Purified human neutrophils have also been shown to retain the IL-6 promoter in a dormant state but to induce expression of IL-6, TLR8 agonist R848, amplified H3K4me3, H4ac, and H3K27ac, those vigorous transcription marks, at the IL-6 promoter after stimulation. The activity of other cells involved in wound healing is epigenetically regulated in immune cells [23]. Similar toJMJD3, standard keratinocyte differentiation is provided by endorsing the expression of genes allied with differentiation, whereasSETD8 and H4K20 methyltransferase props up keratinocyte production and differentiation. Therefore, epigenetic regulators have proven to be a potential therapeutic target for treating a diabetes wound. The various molecular targets and their regulatory mechanisms are described in Table 2.

## 4. Epigenetic Mechanisms and Adaptations in the Diabetic Wound

### 4.1. Modification of DNA

#### 4.1.1. DNA Methylation

Dual mechanisms allow DNA to be modified unswervingly to alter gene transcription: DNA methylation and hydroxy-methylation. The transfer of 5-methyl-cytosine (5mC) from the methyl faction to the cytosine ring of DNA via DNA methyltransferases (DNMTs) is exemplified by the transmission of 5-methyl-cytosine (5mC) from the methyl faction to the cytosine ring of DNA via DNA methyltransferases (DNMTs). De novo methylation marks are deposited by DNMT3A and DNMT3B, whereasthese marks are retained by DNMT1, as these symbols must be recognised after every cell division [24]. In mammals, a gigantic amount of DNA methylation takes place in CpG dinucleotide clusters called CpG coding regions in somatic cells, and about 40 percent of the genes have these islands in their promoters. Numerous tests have shown that methylation of DNA functions in diabetic wounds. 5-aza-cytidine (5-aza-C) inhibition of DNMT1 has also been shown to promote M2-like macrophage development and decrease bone marrow macrophage-derived inflammation (BMDMs). Fascinatingly, these mice were also secluded from obesity-induced inflammation and insulin confrontation. T2D mouse models of genetic (db/db) along with diet-mediated obesity (DIO) have been reported to display increased DNMT1 in BMDMs and to promote a pro-inflammatory macrophage phenotype. Moreover, in db/db mice, DNMT1 knockdown has accelerated wound healing [25]. In vitro polarisation of M2 has been allowed by 5-aza-C treatment. While the rise in wound healing may be attributed to systemic causes because of increased insulin sensitivity and obesity, DNMT1 targeting was recognised as a viable treatment technique in diabetic macrophages and needs more investigation.

#### 4.1.2. DNA Hydroxyl Methylation

DNA methylation has been shown to be irreversible and only occurs when DNMT1 is deficient during replication. However, a dynamic pathway for demethylation of DNA has now been revealed whereby 5mC is chronologically oxidised into 5-hydroxymethylcytosine (5hmC), followed by 5-formylcytosine (5fC), and lastly, 5-carboxylcytosine (5caC) with the enzyme family Ten-Eleven Translocation (TET), which is gradually aloof by the pedestal excision repair method. The moieties 5fC, 5hmC, and 5caC were deliberated to be midway in DNA de-methylation to reinstate gene transcription. However, research has revealed that 5 hmC is uniformly distributed in promoters, gene surfaces, and transcription aspect binding sites in the genome, indicating an independent role in the regulation of gene transcription. It has also been shown that 5fC and 5caC are homogeneously disseminated across the genome, although at diminished levels [26]. Here, is an indication that DNA demethylation plays a significant role in the healing of the diabetic wound. AGE-BSA accumulates due to hyperglycemia and has been shown to inhibit keratinocyte mobilisation and proliferation by fostering TET manifestation that consequently demethylates the MMP9 promoter. TNF increases MMP9 expression in keratinocyte cell lines by de-methylating the site-specific MMP9 promoter.It has also been shown that the targeting mechanism for DNA demethylation could be a successful method for the management of non-healed diabetic wounds [27].

### 4.2. Histone Adaptations

A single histone protein H1 nucleosome linker protein has an octal structure, typically consisting of dual sets of proteins H2B, H2A, H3, and H4. All histone subunits have an N-terminal “tail”, which is separated by an octamer surface, creating an uncovered surface. Depending on the transformations and the modified specific residue, histone-modifying enzymes can either activate or silence transcription [28]. The enzymes that modify histone can also be methylated, acetylated, phosphorylated, or ubiquitylated. These modification in the histone leads to modulation of wound-healing processes such astranscriptional inhibition of genes, inhibition or upregulation of pro-inflammatory cytokines.

#### 4.2.1. Histone Lysine Methylation and De-Methylation

Histone methylation is one of the most studied phenomena in the case of modifications to DNA. Methylation of histone can also encourage transcription or inhibit it based on the targeted deposits and the number of methyl groups involved, unlike DNA methylation, which is often associated with transcriptional repression. Similarly, lysine 4 tri-methylation on histone H3 (H3K4me3) is a typical stimulator of gene transcription, whereas lysine 27 tri-methylation on H3 (H3K27me3) is associated with transcriptional methylation. Although H3K27me3 is related to transcriptional inhibition, H3K27 mono-methylation (H3K27me1) was investigated for transcription promotion. Most notably, these markers are problematic in the polarisation of macrophages, as activated macrophages have been reported to have enlarged methylation of H3K4 and diminished methylation of H3K27 in anti-inflammatory gene promoters. An amplified level of H3K27 demethylase Jmjd3 is responsible for dwindling H3K27 di- and tri-methylation. Methylation of histone H3K4 can be carried outwith separated members of the SET domain-containing protein family. MLL1 has been shown to encourage the NF-dependent expression of inflammatory genes [29]. Diabetic wound healing experiments have shown that monocytes removed from T2D patients undergo augmented MLL1 expression, indicating that during the transition to the propagation process, MLL1 expression is actively regulated. Up until now, this is the only proof that MLL1 has a diabetic wound-healing function. Further analysis is needed to decide whether MLL1 can become a viable therapeutic target. Jmjd3 expression is increased in H3K27me3 histone demethylase and declines in wound macrophages in the DIO model for diabetes. H3K27me3 is an exploitive transcription mark that upholds heterochromatin configuration in gene promoters, whereasJMJD3 promotes transcription. In a DIO diabetes model, JMJD3 has been found to induce IL-12 expression at wound macrophages and to reverse this phenomenon by inhibiting JMJD3. Jmjd3 expression increased in macrophages and has been linked to the regulation of evolved gene expression during bone marrow differentiation in response to inflammatory stimuli [30]. The investigation has shown that in normal neutrophils, epigenetic expression of IL-6 is synchronised in response to TLR activation and is related to increased histone H3K4me3, H3K27ac, and H4 acetylation. However, with less data, more research is needed on the regulation of epigenetics in neutrophils in diabetic wound healing. Research suggests that the expression of Jmjd3 is necessary for re-epithelization. JMJD3 likewise encourages the migration of keratinocytes to the wounded site by fostering Notch1 growth. The study found increased expression of Jmjd3 in epithelial cells at the leading edge of typical wounds, as well as a decrease in Ezh2, the HMT responsible for H3K27 methylation mark deposition. The amount of JMJD3 in wounds was upregulated on day 1 and decreased over time. Thus, in both macrophages and keratinocytes, JMJD3 appears to be closely monitored, where higher expression is crucial in the early stages of wound healing but is downregulated as the wound heals; this regulation appears to be lacking in diabetic wounds [31]. Keratinocyte data relating to JMJD3 and EZH2 have been confined to usual wound conditions. It is necessary to determine their impact on the development of diabetic wounds. At active gene promoters, ASH1L targets histone H3 and is allied with the methylation of K9, K4, K36, and K20 [32]. In typical wounds, the deletion of Ash1l leads to deferred re-epithelialization. However, for ASH1L, no role in the healing of the diabetic wound has been established. In addition, another part of the SET family of HMTs, SET7, is known to be the feature for mono-methylation of histone H3K4 (H3K4me1). In promoter regions, genes are successfully transcribed, and this label is usually located [33]. Hyperglycemia has been shown to boost the fabrication of pro-inflammatory genes and to raise the expression of SET7 in vascular endothelial cells. It is apparent from the study that methylation of histone lysine plays different roles in wound healing depending on the cell types and the modified special histone tail left out. So far, depending on restricted trials, JMJD3 and MLL1 are the most favourable therapeutic targets [34].

#### 4.2.2. Histone Arginine Methylation/De-Methylation

Methylation of histones causes its transcription into arginine metabolites. Here, in the protein arginine, arginine is mono- or methyltransferase-dimethylated (PRMT). In arginine, there are binary exposed amino groups at the terminal, and only one (asymmetric) amino group with two methyls or both (symmetric) mono-methylated groups may be di-methylated. Compared withlysine alteration, there is far less recognition and understanding of the methylation of histone arginine in diabetic lesion healing. Many clinical studies show that PRMTs are involved in the synthesis of insulin in cells and that some are overexpressed in people with diabetes. PRMTs, however, can contain not only methylated histone proteins but also non-histone proteins [35]. It is well known that in diabetic complications, arginine alteration performs a histone-reliant and non-reliant role. There is substantiation that PRMT1 and PRMT4/CARM1 form a complex to control the expression of NFB gene targets with arginine residues of NFB and methylate H3, indicating that arginine methylation may be involved in endorsing the pro-inflammatory responses [36]. However, further research is needed on the methylation of histone arginine in diabetes and especially in diabetic wound healing.

#### 4.2.3. Acetylation/De-Cetylation of Histone

The acetylation of histones is another well-known mechanism. Histone acetyl-transferases (HATs) catalyze the reassigning of an acetyl group commencing acetyl-coenzyme A (Ac-CoA) to the lysine residue terminal amino group lying on the histone, to carry out histone acetylation. Under healthy settings, Lysine positively charged terminal amino groups, employing the negatively charged phosphates in the DNA spine to create a stable combination among DNA and histone. This amino cluster acetylation neutralizes its charge, thereby splitting the communication with DNA and allocating repositioning of the histone and allowing access to transcriptional processes. Acetylation of histone is unique in that it is concomitant solely with the activation of transcription. The above mechanism may also be reversed by extracting histone deacetylases (HDACs) from the acetyl community to avoid gene expression [37]. Ac CoA is a significant by-product formed during the digestion of fatty acids; thus, it seems conceivable that acetylation of histone may have a role in diabetes, and may be involved in the modulation of wound-healing processes in the form of obesity. In reality, the acetylation of histone has been correlated with the cellular metabolic rate, and high-fat diets have been found to affect acetyl-CoA and histone acetylation levels. There are also a few findings on histone acetylation and the healing of diabetic wounds. Global reductions in H3K23 and H3K9 acetylation have been documented in diabetic mouse livers. Acetylated H3K9 is decreased in Glut2 promoters, essential for sustaining homeostasis level of glucose, and may be reversed by treatment with the exendin 4 (diabetic inhibitor). In conjunction with this, a study has shown that exendin-4 can treat diabetic retinopathy by inducing SOD3 promoter acetylation of histone H3 to boost its expression in endothelial cells. Mice displayed enhanced keratinocyte proliferation and better wound healing when treated with HDAC inhibitors [38]. The above research has indicated that histone de-acetylation could be a feasible therapeutic target. However, in diabetes, there is a lack of knowledge on the acetylation of histone and diabetic wound healing.

#### 4.2.4. Other Modifications of Histone

The acetylation of histones is another well-known mechanism. Histone acetyl-transferases (HATs) catalyse the reassignment of an acetyl group from acetyl-coenzyme A (Ac-CoA) to the lysine residue terminal amino group on the histone in order to perform histone acetylation. Under normal conditions, the positively charged terminal amino group of Lysine binds, using the negatively charged phosphates in the DNA spine to form a stable combination of DNA and histone. This acetylation of the amino cluster neutralises its charge, splitting communication with DNA, allocating histone repositioning, and allowing access to transcriptional processes. Acetylation of histone is unique in that it is concomitant solely with the activation of transcription. The above mechanism may also be reversed by extracting histone deacetylases (HDACs) from the acetyl community to avoid gene expression [37]. Since Ac CoA is a significant by-product formed during the digestion of fatty acids, it seems conceivable that acetylation of histone may have a role in diabetes and may be involved in the modulation of wound-healing processes in the form of obesity. In reality, the acetylation of histone has been correlated with cellular metabolic rate, and high-fat diets have been found to affect acetyl-CoA and histone acetylation levels. There are also a few findings on histone acetylation and the healing of diabetic wounds. Global reductions in H3K23 and H3K9 acetylation have been documented in diabetic mouse livers. Acetylated H3K9 is reduced in Glut2 promoters, which is required for maintaining glucose homeostasis, and can be reversed by treatment with exendin-4 (a diabetic inhibitor).In conjunction with this, a study has shown that exendin-4 can treat diabetic retinopathy by inducing SOD3 promoter acetylation of histone H3 to boost its expression in endothelial cells. Mice displayed enhanced keratinocyte proliferation and better wound healing when treated with HDAC inhibitors [38]. The above research has indicated that histone deacetylation could be a feasible therapeutic target. However, there is a lack of knowledge in diabetes about histone acetylation and diabetic wound healing. Histones can also be ubiquitinated and phosphorylated. These indicators have been well investigated and have been assigned roles in transcriptional control. However, no experiments have been performedon wound healing or diabetes linked to these markings, and their functions have yet to be determined.

### 4.3. ATP-Reliant Chromatin Remodeling

Aside from histone acetylation, ATP-hydrolysis-fueled remodelling complexes can also influence histone–DNA interactions. Both ATP-based remodelling complexes contain an SNF2 family ATPase and are classified into two groups based on their subunits, the SW2/SNF12 and the ISWI. The action of each group is similar in that they all use ATP. However, while both naked DNA and nucleosomes can activate SWI2 and SNF2, the ISWI complex must interact with nucleosomes carrying histones through intact amino-terminal tails. Remodeling results are based on the nucleosomal history of the promoter, which results either in the activation or repression of the transcription. While remodelling based on ATP has been widely documented, much of the work in this field has underscored DNA damage and repair in cancer, and comparatively little is known about these processes in diabetes or wound healing [39]. Nevertheless, the enforced growth of cellular ATP can quicken wound healing by encouraging neovascularization and collagen formation. The increased ATP levels caused an increase in BRG1 and BRM ATPases in SWI or SNF intricate components, facilitating a new pathway in anti-inflammatory macrophages, as indicated by the release of MCP-1 chemokines. The deletion of Brg1 impairs the differentiation of keratinocyte terminals and implies a simplified mechanism for ATP-dependent remodelling [40].

## 5. Molecular Targets for Promoting Diabetic Wound Healing

The potential molecular targets are categorised into five factions: immunomodulators, growth factors, neuropeptides, and supplementary agents. In at least one cycle of wound recovery, each of these targets plays an active role. The immunomodulators mainly affect the inflammatory stage but also affect remodelling and angiogenesis. The neuropeptides standardise the provocative stage as well as the angiogenic stage, whereasangiogenesis is controlled predominantly by numerous other factors. Virtually all stages of wound healing, counting hemostasis, aggravation, angiogenesis, and remodeling, are affected by targets. As a result, they are a potential regulator of healing control in the majority of wound phenomena [41].

### 5.1. Immunomodulators

Sequential activation of immune cells, just after inflammatory chemokines and cytokines, occurs throughout the pathological route of wound healing. Cumulative platelets emit platelet-derived growth factor (PDGF) early on, an acknowledged chemokine and an essential immunomodulation for wound healing. The deployment of fibroblasts and neutrophils into the PDGF ascent is a well-known phenomenon. Other inflammatory chemokines, such as interleukins (IL-8 and 6) and macrophage chemotactic protein 1 (MCP-1), are released within 4 to 12 h of wounding when drawn to the wound site and are abundant in neutrophils [42]. There have been studies of novel chemokines, in addition to these well-identified and well-premeditated immunomodulators, whose expression can control the role of immune cells and thus increase wound healing [43].

#### 5.1.1. Lactoferrin

Lactoferrin is a kind of iron-binding protein present in breast milk in significant amounts and, to a lesser degree, in tears, spit, bile fluid, pancreatic fluid, and neutrophil-derived grains. Lactoferrin has been described as a multifaceted immunomodulatory enzyme that regulates the immune system both positively and negatively. First, because of its ability to requisition iron and subvert microbe membranes, lactoferrin has direct antimicrobial properties and is regarded as an important component of host defence derived from mucosal epithelia and neutrophils [44]. As a result, lactoferrin has pro-inflammatory properties that stimulate the release of IL-8, tumour necrosis factor- (TNF-), and nitric oxide (NO) from unstipulated macrophages. Lactoferrin, on the other hand, has anti-inflammatory properties and can inhibit the release of TNF-, IL-1, and 6 beginning endotoxin-mediated immune cells. Lactoferrin has a normalizer function for immune cells in this regard, leading to the initiation of the resting-state immune response while also helping to counteract the overabundant inflammatory response through effective immune cell initiation. Finally, through the upregulation of IL-18 and interferon-g, lactoferrin has been revealed to display antineoplastic assets and has currently shown promise as an adjuvant to chemotherapy in cancer trials [45]. The development of IL-18 in epithelial cells is known to increase lactoferrin, and keratinocytes and Langerhans cells of the skin are an essential dermal supply of IL-18 that may be affected through lactoferrin. IL-18 is known to draw macrophages and neutrophils to the wound site during wound healing, therefore leading to the premature inflammatory process of safe wound repair. Furthermore, it has been shown that lactoferrin drives angiogenesis by specifically inducing the migration and proliferation of endothelial cells, providing a subsequent mechanism by which lactoferrin may be expected to facilitate abrasion healing [46]. Finally, lactoferrin provides an intrinsic benefit for ulceration therapy in that it is a normal endogenous effector contained in elevated concentrations in breast milk and therefore is probable to be effective for either oral or topical administration.

#### 5.1.2. Thymosin β 4

Thymosin 4 (T4) is an omnipresent actin-sequestering fragment that has been identified in multiple cases of diabetic wound healing models. T4 consists of a 43-amino acid protein that was found to be activated during the separation of endothelial cells. A study for the extracellular action of T4 in lesion healing has been recommended as elevated and prominent in wound fluid and platelets, fostering the assumption that T4 is supplied to the damaged tissue from encoded platelets, where it might regulate wound healing [47]. T4 association seemed to be causing a range of beneficial phenotypes in wound cells, including greater in vitro and higher angiogenesis, migration of keratinocytes, deposition of collagen, and in vivo wound contraction. The study reveals that increased T4 expression may have an effect on the structure of metalloproteinase in wounds and wound cells. Metalloproteinase is an important protein for wound debridement, just as ECM and remodeling the collagen are. T4 also performs anti-inflammatory functions by dropping neutrophil immigration in vitro and inflammation-stimulated swelling in vivo [48].

#### 5.1.3. Chemokines in Wound Healing

In advancing wound recovery, the human isoform of chemokines is used. Recently, Human Genome Sciences patented two acknowledged chemokines for use in encouraging wound healing. The first was identified in a stimulation monocyte cDNA library as human chemotactic protein (HCP).As a protein containing119 amino acids with 27 percent distinctiveness and 56 percent resemblance to a human, MCP-1 is predicted to encode the reported cDNA sequence and restrain four cysteine patterns that consist of all chemokines [49]. MCP-1 is a potent monocyte and macrophage attractant developed by the majority of wound-healing cells and has been acknowledged in dermal wounds as an early moderator of severe injury and wound repair. The inventors expect that HCP could confirm functionality in fostering wound treatment, seemingly by reinforcing the initial inflammation, given the similarities to MCP-1. The second alleged chemokine, human chemokine-9 (Ckb-9), was discovered in cDNA records extracted from human breast lymphatic nodule tissue [50]. A 134-amino acid protein with 32 percent distinctiveness and 69 percent resemblance to eotaxin, as well as the four cysteine chemokine motifs, is predicted to encode the reported cDNA sequence. A special chemokine produced from endothelial cells, smooth muscle cells, epithelial cells, and alveolar macrophages, eotaxin has the distinguishing property of being a potent eosinophil chemoattractant. Eotaxin is a remarkable endothelial cell chemokine that has the characteristic property of being an incredible eosinophil chemoattractant. Eotaxin has been implicated in the pathogenesis of inflammatory conditions through eosinophil recruitment and profibrogenic effects on fibroblasts. While eosinophils are currently not known to be important players in the healing of dermal wounds, the inventors believe that Ckb-9 agonists may help recruit cells to remove residues and create connective tissue, whereas Ckb-9 antagonists may help avoid excessive fibrosis [51]. The lack of any laboratory evidence on the biological assets of Ckb-9 and HCP proteins leaves the potential medicinal efficacy of these molecules unclear, whereasCkb-9 and HCP represent innovative research avenues for the improvement of diabetic wound curing technology.

### 5.2. Neuropeptides

The gradual deterioration of both autonomic and somatic nerve fibers characterizes diabetic neuropathy and is the long-term impediment of diabetes. In 30 to 50 percent of diabetic patients, peripheral sensory neuropathy is documented and has been shown to be the most frequent and responsive indicator of foot ulceration [52]. Skin neurobiology has gained a lot of interest lately, and many researchers have shed light on the activity of peripheral nerves in influencing skin progression, such as wound healing. In the blood, signaling linking the central nervous system and peripheral nervous systems contributes to complex immunomodulation in the crosstalk of the endocrine and immune systems. By employing neuromodulators, such as neuropeptides, neurohormones, neurotrophins, and neurotransmitters, this dynamic relationship is mediated. These neuromodulators attach to particular receptors and produce downward signals for skin cell types, involving endothelial cells, fibroblasts, keratinocytes, immune cells, and mast cells. These signaling routes may become compromised in the presence of diabetic neuropathy, leading to the pathobiology of worsened foot ulceration [53].

#### 5.2.1. Substance P (SP)

SP attached to the NK-1 receptor promotes the formation of NO, vasodilation, modified vascular infiltrations, and the assignment and aggregation of leukocytes, encouraging immune responses. SP guarantees the extravasation, immigration, and ensuing aggregation of leukocytes at the injured site, creating a pro-inflammatory microenvironment that further ensures endothelial cell proliferation and angiogenesis [53]. SP functions as an injury healer, assembling MSCs by initiating bone marrow at the injury site, where they participate in wound healing.

#### 5.2.2. Neuropeptide Y (NPY)

NPY release has proven to improve vasoconstriction, smooth muscle vascular multiplication, and cardiomyocyte hypertrophy [54].

### 5.3. Growth Factors

Growth factors are supplemented in each cycle of natural injury repair by platelets, macrophages, endothelial cells, neutrophils, keratinocytes, and fibroblasts, suggesting the importance of growth factors in the repair and healing of wounds. As a result, growth factors have recently sparked a lot of interest in wound management. Becaplermin, a kind of human recombinant PDGF, is the efficient standalone growth factor claimed for diabetic foot ulcer treatment [55].

#### 5.3.1. Vascular Endothelial Growth Factor (VEGF)

Seven members currently make up the VEGF family: VEGF-F, VEGF-A, VEGF-C, VEGF-B, VEGF-E, VEGF-D, and Placental Growth Factor (PlGF); every one of them has a similar VEGF affinity province of eight residues of cysteine that are characteristically spaced. The key molecule implicated in neovascularization and vasculogenesis is VEGF-A. During dermal wound recovery, VEGF has been described as a crucial angiogenesis inducer and is deregulated in diabetic wound treatment [56]. In non-diabetic rabbits, VEGF expression was found to be highly synchronized in wounds placed on mutually ischemic and non-ischemic skin. However, after wound positioning in diabetic mice, VEGF expression was strongly blunted. Furthermore, topical VEGF treatment in models of delayed wound healing has been shown to facilitate wound closure. Management with the angiogenesis enzyme endostatin in mice was revealed to slow wound healing, but this upshot was almost wholly reversed after topical VEGF was added [55]. Topical VEGF application in diabetic mice has been shown to accelerate the repair of cutaneous wounds, in part by organizing and attracting vascular progenitors [57]. While no clinical studies evaluating VEGF in curing diabetic wounds have been reported to date, the completion of such studies is predicted soon.

#### 5.3.2. Fibroblast Growth Factor (FGF)

FGF is a multifunctional entity from the heparin-associated growth factor family that is recognized to facilitate the growth and segregation of a variety of cell types, including dermal fibroblasts, endothelial cells, and keratinocytes. FGF has been concerned with tissue regeneration, wound treatment, and neovascularization, arising from mitogenic and angiogenic functions [58].

#### 5.3.3. Nerve Growth Factor

Nerve growth factor (NGF)-related reactive oxygen species in wound healing involve the recognition of noticeable NGF higher expression of keratinocytes under angiogenic conditions, such as psoriasis and wound healing. In addition, NGF has been revealed to extend leukocyte aggregation, persuade expression of molecules of endothelial cell attachment, endorse the growth and survival of endothelial cells, and improve angiogenesis regulated by VEGF [3].

#### 5.3.4. Connective Tissue Growth Factor (CTGF/CCN2)

CTGF is another factor from the CCN protein family that has been reported to control chondrogenesis, angiogenesis, and fibrogenesis, thereby promoting the wound-healing process. CTGF is a fibroblast derivative that causes fibroblast propagation and ECM [59].

#### 5.3.5. Hepatocyte Growth Factor (HGF)

HGF is liberated through cells of mesodermal origin, which are also referred to as the scatter factor. HGF overexpression resulted in increased granulation tissue and vascularization development in wound healing, as well as indigenous systemic VEGF activation [60].

### 5.4. Other Therapeutic Agents

#### 5.4.1. Homeobox Genes

It is suspected that HoxD3, the chief homeobox gene deliberate in wound treatment, is necessary for typical wound repair and is directly deregulated in diabetic injuries. Preliminary in vitro results showed that HoxD3 in cultured endothelial cells facilitated angiogenesis and collagen synthesis [61]. In endothelial cells adjacent to dermal lesions in normal adult mice, the levels of HoxD3 were then shown to be quickly and stably overexpressed, but not in diabetic mice. In a minor study of human patients, this general pattern of speech seems to have been confirmed, with the additional discovery that the levels of HoxD3 are dampened in diabetic foot ulcerations but not in incurable venous ulcers, indicating a special interaction between HoxD3 and wound treatment, particularly for diabetes and not for other causes of chronic wounds [62]. Finally, after a solitary solicitation of a HoxD3 DNA-permeate methylcellulose blotch, HoxD3 was revealed to facilitate wound healing via the processes of enhanced formation of new blood vessels and collagen accumulation in diabetic mice. Together, these results seem to describe a novel molecular modification specific to the pathogenesis of diabetic wound therapies that can be inverted to further recover the natural reaction of wound restoration.

#### 5.4.2. Treprostinil

Treprostinil, a stable prostacyclin analog sold under the brand name Remodu-lin^®^, is approved for the subcutaneous treatment of pulmonary hypertension [63]. Treprostinil’s application in treating neuropathic diabetic foot ulcers was patented by the drug’s manufacturers in 2005. Prostacyclin is a potent vasodilator that is primarily produced by endothelial cells and stimulates peripheral vasodilation, reduces pulmonary vascular resistance, inhibits platelet aggregation, and inhibits smooth muscle cell augmentation. In diabetics, vascular disease is characterized by disfigurement at both the macrocirculatory and microcirculatory levels. The sole difference between large vessel disease and atherosclerotic alterations in non-diabetics is that diabetics are more likely to develop them in the infrageniculate arteries of the calf. Diabetes microcirculation, on the other hand, is primarily constrained by effective defects in endothelial and smooth muscle cell-arbitrate vasodilation [64]. Therefore, it was hypothesized that prostacyclin’s vasodilatory activities would mostly fail to reverse occlusive macrovascular dysfunction while restoring the diabetic microcirculation’s reduced vascular reactivity. Evidence suggests prostacyclin analogs are poor at encouraging wound healing in the presence of severe coronary artery disease, which is in line with this assumption. In line with this, treprostinil’s creators stress the distinction between neuropathic and neuroischemic ulcers and strongly advocate for the drug’s usage in the former’s therapy [65]. When there is sufficient distal blood supply, neuropathic ulcers—which can develop on the sole or edges—appear without any discomfort. This form of ulcer is more receptive to treatment with the use of a small vessel vasodilator since it is predominantly caused by impaired wound healing due to microcirculatory dysfunction and sensory neuropathy. On the other hand, neuroischemic ulcers are defined as painful toe ulcers that develop in the presence of severe ischemia and reduced distal pulses. Large artery occlusion is the predominant defect causing ulceration in these situations; hence, prostacyclin is not predicted to have a therapeutic impact [17]. Treprostinil has not yet been studied to see if it can help diabetic wounds heal. However, studies involving diabetic mice and the local treatment of an associated prostacyclin analog have demonstrated quicker cutaneous wound healing via improved angiogenesis and blood flow [66].

#### 5.4.3. Nucleic Acid

Captured particulate nucleic acid combines gene therapy and nanotechnology to knock down or prompt a particular gene to treat a severe wound successfully. Gene transmission to the spot of damage facilitates the expression of particular proteins that can accelerate chronic wound healing. For example, in diabetic patients, VEGF for the initiation of angiogenesis in prolonged wounds was transfected by viral vectors, and the result was observed in wound therapy [67]. The use of non-viral vectors such as nanoparticles to supply nucleic acid is, however, a safer option because viral vectors will activate an immune reaction and should therefore be handled with care. SiRNA inhibits gene expression by selectively targeting genes that are overexpressed in chronic wounds, such as MMP, TNF-, and ganglioside-monosialic acid-3 synthases (GM3S).To safeguard against physiological nucleases, in vivo conveyance of siRNA needs a carrier for conveyance into cells. Targeted delivery of siRNA and anticipation of degradationhave been made possible by nanoparticle-based technologies. Clinical experiments using siRNA to treat many infections have been encouraging, but due to insufficient effectiveness or significant off-target results, primary clinical trials were ineffective. Deprivation of siRNAs by enzymes in the wound area and siRNAs not willingly absorbed by the cells due to electrostatic restrictions are obstacles to effective siRNA delivery for efficient therapy, as the negatively charged cell wall would not effortlessly allow the diffusion of negatively charged siRNAs keen on the cells [68].

#### 5.4.4. Antioxidants

The generation of ROS occurs as a result of the recruitment of monocytes, neutrophils, and leucocytes to the wound locations during the inflammatory phase of wound healing. These cells subsequently target microorganisms and foreign debris by phagocytosis. The antioxidant machinery in the cell evolves to play a crucial role in scavenging these free radicals to maintain redox homeostasis, or the balance between free radicals and antioxidants [69]. ROS, which involve the hydroxyl radical, hydrogen peroxide (H_2_O_2_), superoxide (O_2_), and former reactive oxygen derivatives, are highly fatal and cause substantial protein, DNA, and lipid destruction, disrupting the normal functioning of the cells. In encouraging antioxidant behaviors in diabetic rats for successful wound treatment, nanoparticle-dependent therapy has revealed promising performance. Bairagi et al. created PLGA-encapsulated ferulic acid (FA; 4-hydroxy-3-methoxycinnamic acid) nanoparticles to study their effect on diabetic wound healingof the hypoglycemic, neurogenic, angiogenic, free radical scavenging, and antibacterial activity of diabetic wound healing. In this same report, the researchers found that the dispersion of FA-loaded polymeric nanoparticles via oral and topical treatment with hydrogel induced quicker wound epithelization, contributing to successful wound closure on day 14 relative to the community of diabetic wounds [70].

## 6. Pharmacological Approach towards Inflammation

The medication encourages the healing of a wound by influencing multiple phases of tissue regeneration, from inflammation to maturation. The effect of the drug can be enticing or interfere with the specific stage of wound healing progression depending on its mode of operation, dose, and administration route. The following numerous treatment modalities were implemented in the treatment of wound inflammation:

### 6.1. Phyto-Modulators

Plant-based bioactive compounds or supplementary products have the potential to moderate inflammation by substituting cells’ cytokines and growth factors, which are concerned with the treatment of wounds that result in increased angiogenesis, fibroplasia, and epithelization. For their capability to regulate inflammation throughout wound repair, a few promising phytomedicines are briefly mentioned.

#### 6.1.1. Aloe Vera

The mucilaginous gel found in the aloe plant’s leaves has been utilized for its anti-inflammatory and wound healing capabilities since ancient times. The mode of action for wound restoration is allied with suppressing the formation of cytokines, reactive oxygen species, and prostaglandins and increasing fibroblast and keratinocyte proliferation. The carbohydrate content of aloe stimulates the macrophages as well as other immune cells involved in the inflammation phase. The aloe vera treatment improved wound healing in the subsequent-degree burn wound rat model by regulating leukocyte adhesion and cytokine levels, indicating its potential in wound healing [71].

#### 6.1.2. Honey

For their wound-healing assets, various varieties of honey have been used since ancient times. In particular, honey influences the healing process at the wound site by supplying a clammy environment, eliminating bacteria and cell remains, importing oxygen and nutrients, and increasing the propagation of endothelial cells and fibroblasts. Honey’s anti-inflammatory activity is due to the suppression of several factors, such as ROS development, the accompaniment pathway, the penetration of leukocytes, iNOS, COX-2, and MMP-9. Owing to its antibacterial and anti-inflammatory effects, manuka honey has demonstrated promising prospects for wound recovery recently [72]. Honey has been successfully used to accelerate the healing of severe and long-term wounds, indicating its therapeutic benefit in the wound [73].

#### 6.1.3. Curcumin

Curcumin has antioxidant and anti-inflammatory properties liable for faster and improved wound healing, particularly in diabetic rats. Delivery of curcumin by topical route was beneficial in the treatment of both chronic and nasal wounds. Curcumin influences cytokines and growth factors, thus regulating the inflammation mechanism to boost skin healing. It works by decreasing the expression of TNF-, MMP-9, and IL-1 while increasing the expression of the anti-inflammatory cytokine IL-10 and antioxidant enzymes at the wound site [74]. Curcumin-based nanoformulations are proven to accelerate wound healing by inducing fibroblast migration by molecular signaling and inhibiting inflammation by reducing the amount of monocyte chemoattractant protein-1 through fibroblasts.

#### 6.1.4. Picroliv

Picroliv is a normalized portion of the Picrorhiza kurroaa plant’s rhizomes and roots (also known as katuka, kurro, kutaki, or kutki). Picroliv treatment improves endothelial cell sprouting and migration but has also shown enhanced re-epithelialization, angiogenesis, and migration into the wound bed of different cells such as endothelial, dermal myofibroblasts, and fibroblasts. Some of the targets of picroliv, however, may be increased VEGF, reliability with an increased number of microvessels, and increased myofibroblasts in granulation tissue [75]. Picroliv can be used for therapeutic wound healing and angiogenic management in diseases characterized by insufficient blood supply, such as delayed wound healing, and can be used to restore blood supply.

#### 6.1.5. Arnebin-1

Arnebin-1 is a naphthoquinone derivative of the Arnebia nobilis plant root (which belongs to the Boraginaceae genus). The root extract has been extensively used for wound healing in traditional Indian medicine. Arnebin-1 has several biological activities, including wound healing and anti-cancer, anti-bacterial, and anti-fungal effects. At both translational and transcriptional levels, Arnebin-1 improves TGF-1 expression, which is responsible for enhancing wound healing. Arnebin-1 has the impending need for further research as a potent wound-healing therapeutic mediator [76].

### 6.2. Clinical Drugs

#### 6.2.1. Non-Steroidal Anti-Inflammatory Drugs (NSAIDs)

These medications are commonly used in tenderness management. Because of the dissuasion of cyclooxygenases 1 and 2 (COX-1 and COX-2), they retain anti-inflammatory, analgesic, antipyretic, and thrombotic properties. The detrimental impact of NSAIDs on the healing of wounds due to reduced angiogenesis, epithelialization, granulation, and keratinization is demonstrated by various studies. In addition, the 12-HHT/BLT2 alleyway in the skin that affects wound healing may be inhibited by them [77]. However, short-term use of these medications has improved the healing of war-related peripheral wounds by varying inflammatory cytokine levels. Aspirin has been found to have a therapeutic effect on chronic wound healing by inhibiting inflammatory processes and up-regulating repair molecules.

#### 6.2.2. Cyclooxygenase (COX) Inhibitors

These inhibitors impede the manifestation of COX enzymes that are accountable for inflammation-inducing prostaglandin synthesis. Cyclooxygenase has twoisoforms, COX-1 and COX-2, that trigger growth factors, cytokines, and hormones to influence the course of inflammation [78]. In most cells, COX-1 is ubiquitously present, but COX-2 is an inducible form that is found on macrophages, fibroblasts, and other immune cells and is mainly highly expressed in inflammation. To manage the cytotoxic action of inflammation, numerous inhibitors were created for COX-2. Inhibitors of COX-1 have adverse effects on the gastrointestinal mucosa, whereasCOX-2 is safe. After the post-operative procedure, COX inhibitors are used to provide temporary relief from inflammation and permit wound closure. Celecoxib is the most popular FDA-approved COX-2 inhibitor for a variety of family adenomatous polyposis (FAP) and arthritis pains. Celecoxib demonstrated improvement in the animal stimulus ulcer model by inhibiting TGF-1 in prominent wounds to accelerate wound healing and decrease scarring [79]. In a rat mastectomy study, it reduced seroma development and inhibited interleukin-1. In addition, inhibition of the inflammatory reactions, such as neutrophil invasion and initiation, prostaglandin-2 action, and TGF-b1 protein levels, was also shown by the topical delivery of celecoxib to the incision mouse model. Bone marrow mesenchymal stem cells significantly improved wound healing, which was also reported by celecoxib by facilitating re-epithelialization and engraftment [80].

### 6.3. Biological Therapies for Inflammation

At present, more than 20 protein-based recombinant cytokine receptors as well as monoclonal antibody (mAb) medications have been developed and broadly accepted for the treatment of human inflammation (see Table 3). This may be known as cytokine receptor-dependent recombinant proteins or cytokines or cytokine receptor-explicit neutralizing mAbs.

#### 6.3.1. Receptor and Receptor Recombinant Cytokine Receptors-Ig Fusion Proteins

The primary human cytokine-receptor immunoglobulin chimeric combination protein permitted for the management of human illnesses was Etanercept (trade name Enbrel). Etanercept embraces the human TNFR2 extracellular area and the human IgG1 Fc region and is formed in cells of the Chinese hamster ovary (CHO). As a TNFR2-based-Ig protein, it has the properties of both a cytokine receptor and an Ig protein: the TNFR2 portion is attached to TNF and lymphotoxin-, whereasthe human IgG1 portion provides serum durability and binding ability to the Ig Fc receptor (FcR).Thus, etanercept is a kind of TNF inhibitor competent in neutralizing soluble serum LTa and TNF, interacting with and utilizing cytokine-expressing cells (like membrane-bound TNF), and furthermore concurrently interacting with and activating FcR-expressing cells [81]. Similarly, an analogous protein, TNFR1 p55-IgG1 Fc fusion protein (Lenercept), was developed and tested for multiple sclerosis (MS) in a twice-blind, placebo-restricted clinical trial. This choice of disease was based on the fact that TNF is released in MS and has shown cytotoxic activity against oligodendrocytes, the cells killed in MS by the immune system, and since TNF neutralization has been publicized to be effective in mice with autoimmune encephalitis (a murine model for MS-like illness). MS patients, however, showed little benefit from treatment with Lenercept, and, sadly, several patients in the study encountered an unexplained deterioration intheir condition [82]. Clinical trials for sepsis have also failed for Lenercept. The cause for this failure, particularly in the expression of etanercept’s effectiveness, was mysterious at the time and remains mysterious today; it is not clearif the discrepancy in in vivo behavior and therapeutic efficacy is explained by ligand-binding discrepancies or even has a slight variation in the Ig portion. Molecular biological engineering has also produced Onercept, a TNFR1-extracellular section devoid of an FcR portion. In vitro, Onercept neutralized TNF but failed in psoriasis clinical trials [83]. Numerous TNF-inhibitory TNFR-dependent reagents, for example, pegsunercept (a pegylated recombinant soluble TNFR1 protein), have also been developed, but for different reasons, they have not been approved, primarily for lack of effectiveness in the disease circumstances in which they have been experienced. An equivalent recombinant bio-healing IL-1 inhibitor consisting of a distilled blocker protein of the recombinant IL-1 receptor, anakinra, has been formulated and licensed for the management of adult RA, typically given as a weekly subcutaneous (s.c.) injection. In comparison, rilonacept, another IL-1RA (accessory) protein, is a dimeric fusion protein composed of IL-1R1 and an IL-1RA bound to IgG1-Fc. It is approved for the treatment of cryopyrin-linked periodic syndromes, consisting of Muckle–Wells syndrome in adults and children aged 12 and older.

#### 6.3.2. Cytokine-Neutralising mAbs

The earliest anti-human cytokine mAb to be accepted for curative use was infliximab (marketed as Remicade). Human TNF cannot attach to one of its receptors, either TNFR1 or TNFR2, since infliximab binds to both membrane-bound and soluble forms of TNF. Infliximab is a powerful suppressor of TNF’s biological activity, as antibodies are elevated-affinity compounds. Every eight weeks, 5 mg/kg of infliximab is intravenously (i.v.) infused into the patient. There are currently additional therapeutic mAbs that target human TNF. Both adalimumab and golimumab are human and humanized anti-human TNF IgG1 mAbs, sold under the brand names Humira and Simponi, respectively. These mAbs are typically injected subcutaneously (s.c.) once every one to two weeks. An anti-TNF IgG1mAb pegylated human immunoglobulin Fab fragment is known as certolizumab pegol (trade name Cimzia). Additionally, it is given by s.c. injection, typically once per month. All of these medications have been given the go-ahead to treat a variety of human inflammatory disorders linked to arthritic and psoriatic illnesses. Neutralizing IL-1-specific mAbs, such as canakinumab (marketed under the brand Ilaris) and gerokizumab, have also recently been developed (trade name Eyeguard). Both systemic JIA and auto-inflammatory disorders of the CAPS subtype, such as MWS, are approved uses for these. Similar to this, IL-6R-specific blocking mAbs such as tocilizumab (marketed under the name Actemra), sarilumab, and sirukumab have also been created. When methotrexate monotherapy is less effective than anticipated or when anti-TNF drugs are ineffective, these anti-IL-6 mAbs are used in combination with methotrexate to delay the course of RA and JIA in patients. The IL-23 receptor is also blocked by briakinumab, guselkumab, and tildrakizumab. Briakinumab is a human IgG anti-IL-23p40 mAb, and tildrakizumab is a humanized IgG1 anti-IL-23p19 mAb. Both are efficacious and permitted for the treatment of psoriasis. In phase II trials, guselkumab, an IgG1 anti-IL-23p19 mAb, outperformed the anti-TNF mAb adalimumab and was approved for the treatment of psoriasis. It is believed to be secure in earlier-phase trials. These mAbs were just recently created; thus, their safety profiles will need to be continually monitored, even if preliminary data indicate that there is no increased risk of infection [84].

### 6.4. MicroRNAs

#### 6.4.1. miRNA Biogenesis

MiRNAs are trivial endogenous non-coding RNAs with 22 nucleotides that mediate post-transcriptional gene expression regulation. Each miRNA can inhibit hundreds of genes, and the human genome has over 500 of them. MiRNAs are primarily transcribed into primary miRNAs (pri miRNAs) through a stem-loop structure by RNA polymerase II in the nucleus. Nuclear ribonuclease (RNase)-III enzymes smite pri-miRNA through precursor miRNAs (pre-miRNA). Drosha is a non-specific RNase that plays a role in the formation of miRNAs from pri-miRNA; additionally, DGCR8 is a cofactor that aids in the formation of protein complexes. The resultant pre-miRNAs are then transported into the cytoplasm via the nuclear export factors fling-5 [85]. Furthermore, the RNase III enzyme Dicer hews the pre-miRNAs keen on 18 to 24 nucleotides of double-stranded RNA, which then mature into miRNA duplexes. Following Dicer’s cleavage, the outcomes revolve around the 24-nucleotide miRNA duplex required to release single strands for practical entry into Argonaute. To generate RISC-composed miRNA, maximum miRNAs are embedded in the RNA-persuaded silencing complex (RISC). The one filament, also recognised as the passenger strand, usually vanishes, whereasthe additional strand, known as the guide strand, leads the mature miRNA. Nonetheless, recent research suggests that these filaments may be encumbered onto Argonaute and exhibit authoritarianism toward target mRNAs [86]. They accomplish the rationale of amendable gene expression by endorsing mRNA deprivation and suppressing the translation of the miRNA. miRNA duos with the RNA-targeted silencing complex attach to the target mRNA (untranslated) and then inhibit the translation process, which even leads to miRNA degradation [87]. Furthermore, protein-coded genes are synchronised by miRNAs in this approach. The process of biogenesis and functioning of miRNA is presented in Figure 2.

#### 6.4.2. Diverse miRNA in Wound Healing

A variety of experiments suggest that miRNAs have a crucial role in abrasion healing as well as in the polarisation of macrophage regulation. Several miRNAs have been considered crucial at this stage. For instance, miR-146a expression was used to observe an increase in epidermal keratinocyte cell receptor 2 (TLR2), TLR4, TLR3, and TLR5 [54]. It has been shown that miR-146a adversely changes the inflammatory response in intact skin. The downregulation shows that miR-146a can assist in the resolution of inflammation. MiR-155 is a different marker of immune cells that are initiated to be elevated in a mouse model at the inflammatory site, and therapies with selective miR-155 inhibitors can efficiently suppress inflammatory accretion at a wound location and thereby strengthen the architecture of regenerated tissue [88]. In this mechanism, miR-132 shows an anti-inflammatory action by reducing pro-inflammatory cytokine overproduction. Inflammation induces the expression of leukocyte miR-132, and some experiments have shown that its level of expression has shifted throughout the transition from inflammation to the propagation process [89]. Furthermore, miR-21 has been considered important in the resolution of inflammation [90]. There are several miRNA studies available that are linked to polarisation and inflamed reactions, including miR-125b and miR-223. Both in the inflammatory phase and proliferation phase, miR-21 was also found to be crucial. The migration of keratinocytes and fibroblasts is facilitated by miR-21 [91]. Re-epithelization is delayed, and damaged wound contracture is repressed, as a result of an obstruction of miR-21. On the other hand, the available facts prove thesignificant role ofmiR-132 in both inflammation and proliferation [92]. Some findings have indicated that the inhibition of miR-132 would contribute to extreme wound site inflammation, reduced growth of keratinocytes, and delayed wound closure duration [93]. These studies have shown that miR-132 is not only an anti-inflammation factor but also a supporter of the growth of granulation tissue [90]. It has been observed that miR-31 in keratinocytes in the affected region has been substantially up-regulated, and a study has also indicated that miR-31 allows a substantial contribution to migrating and proliferating keratinocytes and to the re-epithelialization process [94]. The downregulation of the miR-99 family has previously been reported to have resulted in the promotion of keratinocyte migration and proliferation, thereby facilitating wound closure. MiR-210 is also found at high levels in hypoxia in ischemic chronic wounds because it is capable of stifling mitochondrial synthesis and minimising oxygen expenditure [95]. Results revealed that rapid wound healing was achieved in a study encapsulating miR-210 inhibitors on lipid nanoparticles and injecting them into murine anguish from an ischemic injury, which highlights the effectiveness of miR-210 in wound healing [96]. Fibroblasts separate into myofibroblasts, collagen deposits, and wound contractions throughout the remodelling process. MiR-29a has been shown to specifically affect the occurrence of collagen, miR-192, miR-29c, and miR-29b, all of which are strongly mediated in this process [97].

#### 6.4.3. Treatment Strategies via Regulating miRNA

As miRNAs are known as capable promoters, they are desirable targets for a wide variety of new restorative techniques (see Table 4). The utmost benefit of miRNA targeting is that the anti-miRNAs or miRNA analogs within the cells could ensure their role even after they are missing from plasma because of their long biological half-lives [98]. Multiple miRNA signals in a chronic wound healing phase were reported to be deregulated. Researchers observed in a current meticulous trial in a model of diabetic rats that, out of the 83 miRNAs, 18 were upregulated and 63 were downregulated, which additionally supported miRNA’s potential for healing wounds. Several studies have shown that miRNA therapy is the frontier in the treatment of various illnesses that eventually result in chronic wounds, including diabetes and a variety of diabetes complications [99].

One of the most difficult problems of wound recovery is the recurrent wound in diabetes, which results in high mortality and amputation. Several RNA-based therapeutics are being developed, including miRNA mimics and suppressors, siRNAs, and antisense oligonucleotides. Therapeutic targeting of miRNA has demonstrated the capability to modulate gene complexes, so effective targeting of miRNA imitators or microvasculature inhibitors would promote cell signaling in damaged tissues that would successively help to improve wound healing. miRNA therapy can also be used to diagnose complications of diabetes that indicate their importance. For example, it was reported that miR-200b expression decreased after an injury to the wound tissue, whereas angiogenesis was found to be altered [100]. In diabetic wound rat replicas, miR-15b and miR-27b playeda key role in the development of new blood vessels and can help substantially improve wound closure [101]. Furthermore, miRNA treatment using anti-miRNA or miRNA mimics would be capable of simultaneously targeting several receptor genes that are cleared from plasma by tissue uptake within hours. Several non-viral miRNA delivery systems have been found to be effective in the wound-healing process, i.e., miR-126 suggests that angiogenesis is an important stage of wound healing. The angiogenic factor, VEGF, is stimulated and blood flow is increased as it is loaded onto polyethylene glycol-modified liposomes and transmitted to an ischemic fibrosis model, which helps facilitate angiogenesis and then wound healing. However, miRNA treatments still have challenges with efficient delivery and targeting. The regulation of miRNAs can be promoted by chemically modified antisense microRNA inhibitors, called antimiRs, or miRNA mimics, but naked antimiRs are not as stable as expected. They deteriorate rapidly in the presence of endogenous transmission and RNases. Therefore, the safety of antimiRs is important to improve target affinity, stability, and efficiency for tissue endorsement [54]. The requirement to administer miRNA to a single type of cell and to maintain sustained target repression with multiple doses are the key problems associated with miRNA replacement technologies. To address these limitations, lipid nanoparticles are used as a transporter for the delivery of oligonucleotide antisense inhibitors to suppress the endogenic messenger RNA (mRNA) in the setting of ischemic wounds, leading to disrupted healing [102]. In common, the local hypoxia-induced miRNAs (hypoxiamir) are increased by an ischemic wound, which denotes hypoxia-responsive miRNA. Potential miRNA treatment targets include hypoxiamiR, as well as miR-203, miR-210, and miR-21, which have proven to be hot candidates as markers of persistent, non-healing wounds that are typically defined in hypoxia. The antihypoxamiR-encapsulated lipid nanoparticles (LNPs) have been developed to enhance the healing of skin wounds and showed a remarkable result from this perspective. Gramicidin A was likewise used to augment endosomal emission and promote the creation of an ion channel in the lipoidal bilayer, utilizing soy phosphatidylcholine (SPC) for lipid bilayer formation, as well as tertiary (DODAP) and quaternary (DOTAP) amine head groups [103]. Intradermal administration of antihypoxamiR LNPs into bipedicle-flap injuries created in mice susceptible to atherosclerosis and diabetes significantly reduced miR-210 expression in the ischaemic wound edge tissue. Preclinical studies also show that the re-epithelialization mechanism was increased after treatment with AFGLNmiR-210 and was followed by apparent epithelial hyperplasia characteristics. Accrued findings have shown that certain miRNAs could be deregulated and participate in multiple diabetic wound-healing pathophysiological processes. For example, in diabetic conditions, miR-27b is overexpressed, which increases cell accumulation, tube formation, adhesion, and inhibits apoptosis. The successful targeting of TSP1-2, semaphorin 6A, and p66shc (Src homology 2 domain-containing transforming protein 1) in angiogenic cells (BMAC) arising from the bone marrow may achieve all these therapeutic goals. It is noted that overexpression of miR-27b contributed to better wound healing and wound perfusion after topical delivery of BMACs in diabetic mice. Wang et al. observed that direct miR-27b delivery could only partially improve wound healing, signifying that this method could also include other miRNAs [101]. The miR-99 family is lowered in diabetic wounds, accompanied by PI3K/Akt signaling pathway interference and keratinocyte proliferation. The MiR-99 family also plays a vital function in re-epithelialization [104]. In addition, miR-155 effectively regulates the immune response and is very effective in diabetic mice. MiR-155 demonstrated its utility through the specific genes SHIP1, RhoA, BCL6, and FIZZ1. MiR-155 insufficiency contributes to reduced inflammation and increased wound closure [103]. In addition, miRNA-26a has also been reported to have caused a diabetic dermal injury in the mouse model. When miR-26a expression was inhibited, multiple wound recovery events, including increased granulation tissue and angiogenesis, were observed by modifying the signaling pathway D1/SMAD1. Nevertheless, miR-26, a local inhibitor, did not affect leukocyte accretion and enhanced myofibroblast deposition [105]. These studies collectively demonstrated significant impacts of dermal fibroblast activity from the microvasculature on diabetic wound healing. Gene therapy for chronic wounds may be feasible based on the recognition of microRNAs, but challenges exist with effective delivery processes to extracellular and intracellular tissues. The key drawback is that only one miRNA silences several proteins, making the consistency of the procedure the problem. In addition, dosage issues should be resolved before incorporation into clinical practice, as miRNA toxicity may lead to non-specific immune reactions and toxicities. Furthermore, standard miRNA applications are preferred at sufficient in vivo physiological levels [106].

**Table 4 biomedicines-11-00613-t004:** The significance of miRNAs in numerous phases of wound healing along with their potential therapeutic targets.

Phase	miRNAs	Targets	Functions	Reference
Inflammation	miR-146a	TRAF6, IRAK1	inhabit excessive inflammatory reactions in keratinocytes and macrophages	[106]
	miR-155	BCL6, SHIP1	control development and activity of immune cells	[104]
	miR-132	HBEGF	Increases anti-inflammatory transmitter acetylcholine level reduces chemokine release by keratinocytes	[93]
	miR-21	PTEN, PDCD4	Suppress LPS-induced inflammatory responses	[84]
	miR-125b	TNF-α	divergence in regulation	[107]
	miR-223	Mef2c	Polarization regulation	[108]
Proliferation	miR-21	TIMP3, TIAM1	Promotes migration of keratinocytes and fibroblasts	[105]
	miR-132	HBEGF, RASA1	Induce keratinocyte growth and neovascularization	[93]
	miR-31	EMP1	Promotes propagation and immigration of keratinocytes	[84]
	miR-99 groups	IGF1R, AKT1,mTOR	Silences keratinocytes migration and proliferation	[101]
Remodeling	miR-210	E2F3, EFNA3	Suppresses keratinocytes proliferation promotes angiogenesis	[99]
	miR-29a	Collagen I and II	Improves collagen expression	[109]
	miR-29b	COL1, COL2, COL3A1	Develop ECM remodeling	[110]
	miR-29c	SMADs, β-Catenin	Progress remodeling of * ECM	[111]
	miR-192	E-Catherin	Enhance * ECM modification	[112]

* ECM—extracellular matrix.

### 6.5. Bone-Marrow-Derived Mesenchymal Stem Cells (BMMSCs)

Mesenchymal stromal cells display a high prospective for wound healing in the skin because of their demarcation, quick processing, and decreased immunogenicity. MSCs also emerged as potential management options for acute and chronic wounds due to their multilinear separation, pro-angiogenic, and immunomodulatory properties. They provide faster healing through increased cell division, angiogenesis, epithelialization, and granulation tissue growth. In addition, chemoattractants for macrophages and keratinocytes are secreted. Improved wound healing has been reported in preclinical and clinical trials of MSCs. Intraperitoneal and intra-regional administration of MSCs showed fast healing in full-thickness wound models in mice. Topical BMCs applied by fibrin spray to chronic low-extremity wounds and acute surgical wounds significantly reduced the wound. Additionally, studies showed that non-obese diabetic mice treated with mouse-bone-marrow-derived allogeneic MSCs had improved skin regeneration and wound healing compared with animals treated with acellular derivatives [113].

### 6.6. Medical Maggot Therapy

Maggot debridement therapy (MBT) is known to use live, disinfected maggots inserted in non-healing abrasions to sterilize the wounds and remove the wound’s necrotic tissue. Experiments have shown that maggot excretions and emissions can decrease the release of superoxide while also reducing myeloperoxidase (MPO) release from activated neutrophils. MBT has wound-healing, debridement, stimulator, and antibacterial properties, as well as effects on inflammation, angiogenesis, and cell proliferation [114]. It promotes healing in diabetic foot ulcers by modifying monocyte function by decreasing pro-inflammatory cytokines (TNF-, IL-12p40, and MIF) while increasing anti-inflammatory cytokines (IL-1).

### 6.7. Fluorescence Bio Modulation

It is a type of laser treatment where red and proximate-infrared lights are implemented to reduce inflammation and promote wound healing. FBM’s anti-inflammatory action has been observed in different animal models, such as acute traumatic brain injury, autoimmune encephalomyelitis, and lung inflammation. The super-pulsed laser successfully reduced inflammation in the rat burn wound model by decreasing NFkB and TNF- and increasing levels of VEGF, HSP-90, FGFR-1, HIF-1, HSP-60, matrix metalloproteinases (MMP)-2 and MMP-9, thereby improving wound condition [115]. A multi-centered, potential, and unrestricted clinical trial of prolonged ulcer patients undergoing FB therapy exhibited remarkable improvement. FB tends to be an attractive alternative for the management of acute and chronic wounds. FB pathways to increase wound restoration include the decline of microbial contents, a decrease inpro-inflammatory cytokines, cell proliferation activation, angiogenesis enhancement, better collagen fabrication, and least scar development [116].

### 6.8. Hyperbaric Oxygen Healing

Oxygen deficiency tends to obstruct the healing processes of wounds. It is reported that immune cells, inflammatory cytokines, and bactericidal mediators are stimulated and modulated in such cases. Hyperbaric oxygen therapy (HBOT) refers to the therapeutic use of pure oxygen at elevated compression and is used in chronic non-healing wound conditions such as diabetes. It is primarily useful for such long-term wounds with low tissue blood flow and oxygen availability. HBOT has been revealed to regenerate muscle damage through oxygenation, decreased inflammation, and regeneration through the activation of macrophages and microglia. Through the enhancement of wound treatment by rising levels of VEGF and IL-6 and downregulation of endothelin-1, this therapy also restored chronic wounds in non-healing conditions. HBOT demonstrated inhibition of microorganism development in soft tissue and bone infections, thereby enhancing macrophage and leukocyte activity for successful wound restoration [117].

## 7. Novel Treatment Interventions and Drug Delivery Mechanisms

It is apparent that these pathways epitomize favorable targets for potential therapeutic therapies focused on the above-mentioned pathways influencing skin repair and fibrosis. The skin has tremendous potential for uninterrupted application of medications and cell therapies because of its approachability, but its activity as a buffer impedes the absorption of such drugs, which it interprets as mainly ineffective. Penetration of the stratum corneum into the skin at risk of ulceration, preservation of cell viability after transmission, and the production of successful continuous delivery mechanisms are some of the difficulties that still cause problems with effective localized delivery to the skin. The routes of permeation of the drug through the skin in the vascular area of the dermis are narrowly described as (1) the transcellular pathway (corneocyte transmission); (2) the intercellular pathway (lipid matrix diffusion); and (3) the shunt pathway or appendage pathway (diffusion into the sweat gland, hair follicles, and sebaceous gland). The medication enters the low-lipid regions of the transcellular pathway in the cytoplasm of the corneocytes, whereasthe intercellular lipid matrix consists of dense lipids such as ceramides and fatty acids. This pathway is also shared by both hydrophilic and lipophilic regions, making this route very immune to drug permeation. Permeated molecules and nanoparticles in the shunt pathway migrate via sweat ducts and hair follicles. The exterior area is protected by the appendages, which are merely around 0.1 percent of the skin’s spectrum; this channel is thus known as a smaller route of transdermal infusion. The nanoparticles can seep through the appendage and act as reservoirs for the localized delivery of skin wound healing medicines [118].

### 7.1. Lipoidal and Polymeric Nanocarriers Formulation for Efficient Skin Penetration

Encapsulating topical drugs in nanoparticles or cationic liposomes, which are tiny, bi-layered lipid particles that may have effortless penetration of the skin barrier, increases the skin’s receptivity for topical treatments. In this way, an optimum condition for repair rather than fibrosis can be achieved by adding vitamins (mostly retinoids), antioxidants (for example, curcumin), and antibacterial substances. In addition, acellular hydrogels comprising either chitosan or dextran polymers have also revealed tremendous promise, in addition to artificial skin dressings that imitate the functions of the skin and are cosmetically attractive. For example, topical delivery of p53 siRNA molecules uniformly dispersed in an agarose matrix shows better cessation of ulcers linked to diabetes by up-regulation of the vasculogenic signaling pathway [119].

### 7.2. Prevention of Malicious Stress-Induced Wound Healing

Signals and stress-related hormones can reduce viable healing of injuries by up to 40%. Therefore, improved wound control can be achieved by enhancing the fibrotic healing process by alleviating these stressor pathways. Catecholamines are also liberated in stress, in addition to steroids, and while beta-blockers can inhibit these molecules directly, they symbolize a potential addition to wound healing medications, as seen through the arsenal management of propranolol in the treatment of childhood hemangiomas [120].

### 7.3. Noteworthy Mode of Topical Gene Therapy

The aforementioned attempts to directly deliver the wound-encouraging genes into the site with exposed DNA molecules were mostly ineffective because of a cutaneous barrier to the skin. Gene-encapsulating viral and non-viral vectors have been used to bypass the low transfection rate. Adeno-related vectors can be utilized in gene therapy for the development of repair-promoting genes such as VEGF, which have had excellent outcomes in topical application to improve angiogenesis and circulating cell recruitment that originate from the bone marrow at the wound spot. In contrast, a retrovirus can be conscripted to carry genes in a similar manner and has shown positive results in both in vitro and in vivo regeneration of skin barriers. Non-viral genetical vectors can also be used to avoid infectious problems caused by virus particles, including liposomal carriers. Liposomes with dual-phase architecture, as previously described, along with the presence of an active pharmaceutical agent in their hydrophilic interior, also have genetically engineered substantial, whereas their small size and hydrophobic external membrane allow direct transportation of this substance into the tissue [68]. The vectors have demonstrated outstanding effects in skin permeability and transfection, with or without viral gene therapy. A powerful wound healing technique was developed explicitly with the Fidgetin-Like 2 (FL2) enzyme by way of in vitro research using FL2-diminished mammalian cells, suggesting its activity in preventing cellular movement and re-epithelization. In addition to these outcomes, the enzyme has been more effective in preventing wound replication than any other cell. Further in vivo studies in wound-bearing mice with the topical implications of FL2 siRNA nanoparticles culminated in a marked improvement in healing acceleration and wound closure [52].

### 7.4. Stem Cells Initiate Fibrotic-Free Treatments

Meanwhile, since the stem cell reservoir is the primary source of repair for the skin, it too signifies a perfect target to trigger a latent curative process. This method of cell-dependent therapy can dwell on several forms of stem cells, including mesenchymal stem cells derived from bone marrow, adipose, and cutaneous tissues. When included in fibrillin or collagen skin dressings, stem cells from adipose tissue and bone marrow have great potential to promote cutaneous healing because they can inhibit growth factors that target the stem cell niche of the skin. Another method of employing skin stem cells and altering them to produce a scar-free reaction rather than a fibrotic one is the transplantation of autologous cutaneous stem cells to the wound site. Interestingly, the hair follicle is an abundant reservoir of these cells and harbors an elevated number of stem cells that can differentiate into skin-regenerating cells, aside from hair fibers [120].

### 7.5. Biomaterials-Based Therapy

Since the earliest texts on medicine, biomaterials have been employed to speed up wound healing. Ancient Egyptian literature mentions the use of honey, grease, and vegetable fiber to speed up the healing of wounds. After World War II, the use of new polymers has been restricted for medical devices. This includes nylons, fluoropolymers, silicones, methacrylates, and polyesters. They are implemented for biomedical therapies such as vascular grafts, hip prostheses, hydrocephalus shunts, intraocular lenses, kidney dialysis systems, and other former medical devices. Currently, the three main kinds of biomaterials employed in the medical field are ceramics (orthopedic and oral implants), metals (stents, dental implants), and polymers (sutures, joint tissue, vascular grafts, and soft tissue in general). Particularly for wound dressings, encapsulation of cells, and nanoparticles, biomaterials have proven crucial to the wound care sector. Biomaterials used in wound healing can be stand-alone with bioactive components, cell encapsulating, nucleic acid delivering, animal product-based, drug- or antibiotic-loaded, or a combination of these. They consist of different biomaterials, i.e., standalone urethane, collagen, and siloxane, with bioactive agents made up of natural components such asfibrin, hyaluronic acid, and many more. They are used in wound healing depending on the constituents of the biomaterial.Due to their adaptable characteristics and release kinetics, biomaterial-based wound dressings are perfect for loading medications or antibiotics. Drug-loaded biomaterials are formulated with chitosan, PEG, carrageenan, dextran, and polyol, which act as linkers, reduce oxidation and microbial contamination, and untimely regulate wound healing. The biomaterials used to encapsulate the cell are made of fibrins and poly amino ester, which promote angiogenesis and tissue regeneration [121].

## 8. Conclusions

Wound healing is an intricate and strictly regulated mechanism involving multisystem innervation and the working of various kinds of cells. Diabetic wounds are characterized by recurrent immune cell changes that leave unhealed wounds behind. a sequence of well-harmonized cellular progressions resulting in the healing of wounds affected by the inflammatory response. Heavy inflammation is responsible for poor wound treatment in chronic wounds. The requirement for effective and effective care for wound healing is extremely critical. Attempts are required to build technologies that facilitate the complete healing of wounds. The current review examines the role of different cellular and molecular targets for inflammation in wound healing and their focus on tissue repair. Studies also demonstrated the essential roles of different therapeutic targets in different facets of chronic diabetes-related wounds, including neuropeptides, substance P, T cell immune response cDNA 7, lactoferrin, homeobox genes, hepatocyte growth factor, miRNA, and treprostinil growth factors such asfibroblast growth factor, including thymosin beta 4, and immunomodulators. These molecular activities offer an alluring clinical approach to better strengthen the mechanism of wound healing. By modulating the activity of these molecular and cellular targets, it is possible to focus on a variety of nearly identical targets in a mechanism responsible for slowing wound healing, which is highly feasible relative to conventional drug treatment. New therapies would aim to achieve a combination of the cascade of inflammatory infiltrate healing by reducing the negative effects of inflammation while retaining the beneficial features that can promote improved tissue repair. Furthermore, the configuration of the efficient delivery mechanism must overcome several extracellular and intercellular obstacles to ensure that the loaded medicaments are efficiently transmitted to the stated target while avoiding several immune reactions.

## Figures and Tables

**Figure 1 biomedicines-11-00613-f001:**
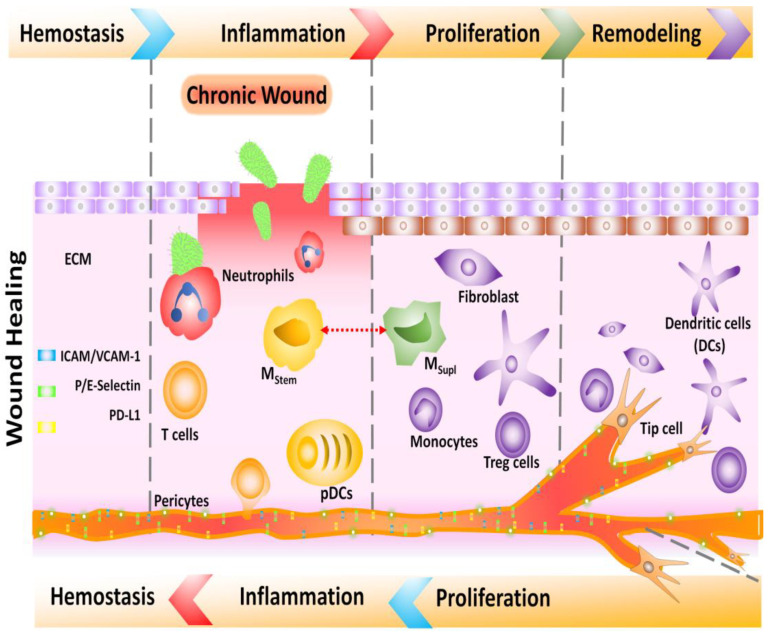
Mechanism of wound healing: The figure depicts the following: 1. hemostasis: reduced blood flow, platelet aggregation, inflammation initiation; 2. Inflammation: neutrophil infiltration, monocyte recruitment, macrophage differentiation, lymphocyte infiltration; 3. Proliferation: re-epithelization, granulation tissue formation; 4. Remodeling: collagen remodeling, apoptosis of fibroblasts, reduction inthe scar.

**Figure 2 biomedicines-11-00613-f002:**
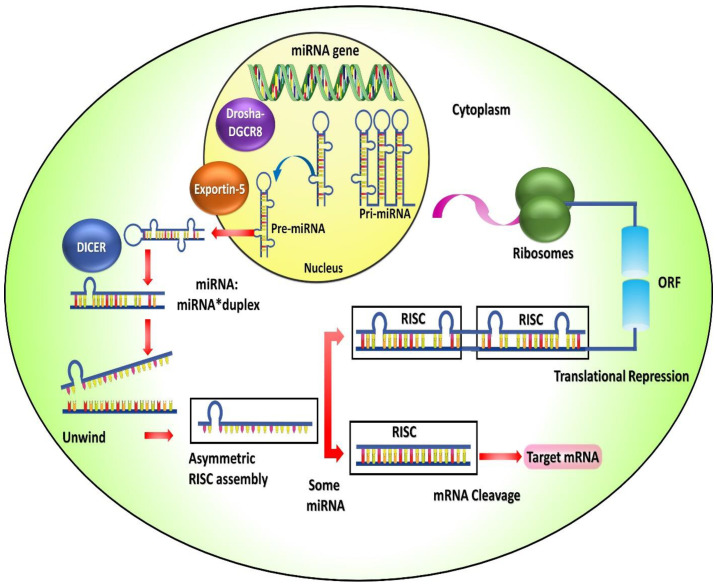
Pri-miRNA transcription, which occurs in the nucleus, is the first step in the biogenesis of miRNA. The pri-miRNA is broken down via the Drosha and DGCR8 complex to create pre-miRNA, which can subsequently be exported by exportin 5 (XPO5) as of the nucleus into the cytoplasm. Dicer continues to transform pre-miRNA in the cytoplasm into the mature miRNA duplex. When a mature miRNA attaches to the 30 UTR of its aim to mRNA, it forms a complex with Dicer and the Argonaute protein known as the miRNA-enclosed RISC. This suppresses or degrades the target mRNA’s translation.

**Table 1 biomedicines-11-00613-t001:** Growth factors and cells’ roles in wound healing.

Cell Growth Factors	Function in the Healing of a Wound
VEGF	Formation of blood vessels in granulation tissue.
FGFs	The abundance of epithelial cells and fibroblasts, matrix deposition, and angiogenesis.
KGF’s	keratinocytes migration and its proliferation
EGF	Differentiation, immigration, proliferation, development of granulation tissue, adhesion of keratinocytes
G-CSF	Encourage the fabrication of neutrophils, increases the utility of neutrophils and monocytes, and stimulates keratinocytes proliferation.
PDGF	Myogenic for smooth muscle cells and endothelial cells Chemo-attractant for neutrophils and Fibroblasts, Collagen metabolism and proliferation of fibroblast.
GM-CSF	Facilitates the propagation of epidermal cells.
TGF-α	Stimulates accumulation of fibroblast and epithelial cells, and development of granulation tissue.
TGF-β	Myogenic for smooth muscle cells and fibroblasts, macrophages Chemotactic, excite metabolism of collagen, and angiogenesis (indirect).
IL-1	Neutrophil chemotaxis, a proliferation of fibroblast.
TNF	Fibroblast proliferation.
Monocytes (macrophages)	Phagocytosis and wiping out invading bacteria, clearance of debris, and necrotic tissue, abundance of inflammatory mediators consisting of cytokines, stimulate the division of fibroblast, angiogenesis, and collagen synthesis. Time of action—8 h.
HGF	Neo-vascularization, Re epithelization.
IGF-1	Stimulates proliferation of Fibroblast. Synthesis of sulfated proteoglycans and collagen
Serotonin	Vasoconstriction
PGE2	Disaggregation of platelet, pain, fever.
Leukotrienes	Chemo taxis and increased vascular permeability.
Lipoxins	Weaken inflammatory response.
Interferon	Maturation of macrophage along with the release of Nitric Oxide.
Platelets	Activates coagulation cascade and involves in thrombus formation. Releases mediators of inflammation. Time of action—seconds
Neutrophils	Phagocytosis of invading bacteria, debridement of wounds along with the release of proteolytic enzymes. Time of action—Peak at 24 h
Lymphocytes	Unknown mode of action. Produce cytokines and regulate the wound healing proliferative phase. Time of action—72–120 h
Fibroblasts	Fabricate components of the extracellular matrix, synthesis of collagen, and granulation tissue. Time of action—120 h

EGF: Epidermal growth factor; G-CSF: Granulocyte; FGF: Fibroblast growth factor; HGF: Hepatocyte growth factor; IGF-1: Insulin-like growth factor-1; IL-1: Interleukin-1; KGFs: Keratinocyte growth factors; TNF: Tumor necrosis factor; MMP: matrix metalloproteinase; PDGF: Platelet-derived growth factor; TGF-α: Transforming growth factor-α; PGE2: Prostaglandin E2; TGF-β: Transforming growth factor-β; VEGF: Vascular endothelial growth factor.

**Table 2 biomedicines-11-00613-t002:** Mechanisms of molecular regulation in the wound-healing process.

Molecular Event	Action	Molecular Mechanism	Activity in Wound Healing
Ca^2+^	Transcription-self-governing diffusible damage signals	Tissue injury causes a prompt rise in Ca^2+^ intracellular, that modify gene transcription via a protein kinase C and Ca^2+^/calmodulin-dependent protein kinase (CaMK)	Elevation of actin polymerization and actomyosin contractility of fibroblast and keratinocytes Increased actin dynamics Improved expression of wound response genes
H_2_O_2_	Transcription-independent diffusible damage signals	Concerned in the establishment of chemotactic signals that vigilant the immune system to damage	Modulation of hemostasis, inflammation, proliferation, angiogenesis, epithelialization, and remodeling phases of wound healing
ATP	Transcription-autonomous diffusible damage signals	A mechanical injury arises a rapid and substantial ATP liberation by dented cells into the extracellular space	Stimulation of the wound healing cascade
miR-146	miR	commencement—epigenetic signal	NFκB Activation directive of innate immune responses
miR-221 and 222	miR	Trigger—epigenetic signal	Angiogenesis
miR-125b	miR	Reticence—epigenetic signal	suppress regulation of TNFα inflammatory genes
miR-210	miR	Activation—epigenetic signal	embarrassment/activation of keratinocytes propagation
miR-146a	miR	Stimulate—epigenetic signal	Produce ECM proteins in chronic diabetes complications
miR-27b	miR	provoke—epigenetic signal	Activation of cell proliferation and adhesion Inhibition of oxidative stress responses Improvement of new vessel formation
miR-203	miR	Activation—epigenetic signal	Activation of keratinocytes propagation
Metabolic memory	DNA methylation	Epigenetic signal	Diabetic foot fibroblasts and ulcers had lesser global DNA methylation contrast by non-diabetic foot fibroblasts
Polycomb Group (PcG) class of genes	Chromatin gene repression	Epigenetic signal	Downregulation of threerepressive PcG proteins (Eed, Ezh2, and Suz12) thru wound healing
Trithorax group (trxG) class of genes	Chromatin gene activation	Epigenetic signal	Up-regulation of twoactivating trxG members (Jmjd3 and Utx) within wound healing
Quantitative trait loci (QTL)	Controlled intuitivetrait	entity unpredictability of the gene expression	Gene expression dissent stimuli the rate and instance of wound healing effectiveness
Fibronectin (pFN and cFN)	Gene polymorphisms	Alternative interweaving	The splicing based on type of cell, its function, and the phases of development
Poly (ADP-ribose) polymerase (PARP) enzymes	PARylation	PARPs amended nicotinamide commencing NAD+ and affix the enduring ADP-ribose entities to suitable protein acceptors. DNA reparation active PARP enzymes	Accelerate wound closure and keratinocytes migration. Prompt synthesis of inflammatory mediators and the wound repairing activity of keratinocytes
MtROS	Mitochondrial Reactive Oxygen Species	Fabricate Reactive oxygen species (ROS) in mitochondria	Promote actin-centered epithelial wound healing Anti-bacterial action Regulate migration of endothelial cells

**Table 3 biomedicines-11-00613-t003:** Currently available biological therapeutic agents.

Drug/Biologics as Therapeutic	Nature/Structural Composition of Protein	Brand/Company	Location
Etanercept	Recombinant fusion protein: Human Tumor necrosis factor receptors 2: ImmunoglobulinG1-Fc	Enbrel^®^	China
Rilonacept	Recombinant IL-1R accessory protein (*E. coli*-derived)	Arcalyst^®^ (Regenron Pharmaceuticals)	Eastview, Mount Pleasant, New York, USA
Adalimumab	Human ImmunoglobulinG1κ	Humira^®^	Puerto Rico
BI655066	Human mAb anti-IL-12/IL-23 p40 ImmunoglobulinG1	(Boehringer Ingelheim Pharmaceuticals)	Ingelheim, Germany
Golimumab	Human ImmunoglobulinG1κ	Simponi^®^ (Janssen Medica)	Belgium
Certolizumab Pegol	Pegylated-Fab’ of humanized ImmunoglobulinG1κ	Cimzia^®^ (Union Chimique Belge)	Belgium
Erelzi	TNFR2-IgG1 Etanercept bio similar	etanercept-szzs^®^ (Sandoz)	Holzkirchen, in Germany
CTP-13	Humanized Immunoglobulin G1κ Infliximab biosimilar	Remsima^®^ (Infliximab) Inflectra^®^ (Hospira)	United States, America
Brenzys (SB4)	TNFR2- Immunoglobulin G1 Etanercept biosimilar	(Samsung Bioepis; Merck and Biogen)	United states of America
BOW015	Human IgG1κ Infliximab bio similar	Infimab^®^ (Reliance Life Sciences)	India
SB2	Human ImmunoglobulinG1κ Infliximab bio similar	(Samsung Bioepis; Merck and Biogen)	United states of America
Adalimumab-atto	Human ImmunoglobulinG1κ Adalimumab bio similar	Amjevita^®^ (AMGEN)	America
Adalimumab (India)	Human IgG1κ Adalimumab bio similar	Adfrar^®^ (Torrent Pharma)	United states of America
SB5	Human IgG1κ Adalimumab bio similar	(Samsung Bioepis; Merck and Biogen)	United states of America
Infliximab	Humanized (chimeric) ImmunoglobulinG1κ	Remicade^®^	Beerse, Belgium,
Anakinra	Recombinant human IL-1Rα (protein derived from *E. coli*; non-mAb)	Kineret^®^ (AMGEN/ Biovitrum)	America
Gerokizumab	Humanized mouse anti-human IL-1β ImmunoglobulinG2κ (Fab)	EyeguardTM (XOMA Corp.)	Emeryville, California,
Canakinumab	Humanized anti-IL-1β ImmunoglobulinG1κ	IlarisTM (ACZ885) (Novartis)	Switzerland
Sirukumab	Human mAb ImmunoglobulinG1κ	(GlaxoSmithKline)	London, England
Tocilizumab	Humanized mouse anti-IL-6R ImmunoglobulinG1κ	Actemra^®^ (Hoffmann–La Roche)	Basel, Switzerland.
Brodalumab	Human anti-IL-17R IgG2κ	(KHK4827, AMG827) (Valeant Pharmaceutical & Kyowa Hakko Kirin)	Tokyo, Japan
Sarilumab	Human anti-IL-6R IgG1κ	VelocImmune^®^ (Sanofi and Regeneron)	New York, USA
Tildrakizumab	Humanized mAb Anti-IL-23 p19 IgG1κ	(Merck; and now Sun Pharma)	India
Ixekizumab	Humanized anti-IL-17A and17A/F ImmunoglobulinG4	Taltz^®^ (LY2439821 Eli Lily & Co.)	America
Secukinumab	Human anti-I7A ImmunoglobulinG1κ	Cosentyx^®^ (Novartis Pharma AG)	Switzerland
Ustekinumab	Humanized mAb anti-IL-12/IL-23 p40 ImmunoglobulinG1κ	Stelara^®^ Jassen-Cilag andCentocor	Beerse, Belgium
Briakinuman	Human mAb anti-IL-12/IL-23 p40 ImmunoglobulinG1κ	ABT-874 (Abbott)	Ravenswood, Chicago
Guselkumab	Humanised mAb Anti-IL-23 p19 ImmunoglobulinG1κ	(Janssen Research & Development)	Netherlands.
AMG139	Human mAb p40 IgG1κ anti-IL-12/IL-23	(Amgen)	Oaks, California

## Data Availability

Not applicable.

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
