# Peer review of "Novel Biotherapeutics Targeting Biomolecular and Cellular Approaches in Diabetic Wound Healing"

_biomedicines, 2023, doi:10.3390/biomedicines11020613_

Round 1

Reviewer 1 Report

1) There are a lot of incomplete sentences and typos. For eg: Abstract (Line 17-18). Also, the first sentence of the introduction (Line 28-29) is very non-scientific. The authors should consider fixing all of these typos for clearer reading of the manuscript. 

2) The manuscript adds limited insight into the field of wound healing that is already not been covered elsewhere. Also, most of the information is written in a more general format, although it was intended to be written in the context of diabetic wound healing. 

3) There is very little background on the complex wound healing process in diabetes, how healing is dysregulated in a diabetic wound, and how it compares to a healthy wound. There is also very little information on the comorbidities that can affect wound healing in diabetes patients. The pathophysiologic relationship between diabetes and impaired healing is complex. Vascular, neuropathic, immune function, and biochemical abnormalities each contribute to the altered tissue repair. 

4) The headings for each section are very poorly chosen and in some cases very obvious. ("Skin is an optimal organ for illuminating pathways for skin repair";) 

4) This review is written more as a textbook than as a critical review of the research performed in the field. 

5) A big portion of skin wound healing is biomaterials based therapy. This is a >50% portion of the research and the authors have not addressed this. 

Author Response

Reviewer 1

Comments and Suggestions for Authors

  • There are a lot of incomplete sentences and typos. For eg: Abstract (Line 17-18). Also, the first sentence of the introduction (Line 28-29) is very non-scientific. The authors should consider fixing all of these typos for clearer reading of the manuscript. 

Reply 1:- We have correct all the typographical error in the manuscript and fix the non-scientific sentences. All the corrections are done by using track changer.

  • The manuscript adds limited insight into the field of wound healing that is already not been covered elsewhere. Also, most of the information is written in a more general format, although it was intended to be written in the context of diabetic wound healing. 

Reply 2:- We have revised our content in more scientific way in the context of diabetic wound healing.

  • There is very little background on the complex wound healing process in diabetes, how healing is dysregulated in a diabetic wound, and how it compares to a healthy wound. There is also very little information on the comorbidities that can affect wound healing in diabetes patients. The pathophysiologic relationship between diabetes and impaired healing is complex. Vascular, neuropathic, immune function, and biochemical abnormalities each contribute to the altered tissue repair. 

Reply 3:- The dysregulation and the factor influence the in the diabetic wound healing is explained in the section 3 entitled Diabetic foot ulceration and delayed wound healing/ Mode of impaired diabetes wound healing. Further the complications in the diabetic wound healing has been elaborated.

  • The headings for each section are very poorly chosen and in some cases very obvious. ("Skin is an optimal organ for illuminating pathways for skin repair";) 

Reply 4:- Some of the heading and subheadings of the review articles has been modified according to the content.

  • This review is written more as a textbook than as a critical review of the research performed in the field. 

Reply5:- In these review paper we have covered the researched from last 10 year. Further, after the reviewer comments the content of the paper has been improved. In feature these paper is very helpful in research and reviewing the diabetic wound healing.  

  • A big portion of skin wound healing is biomaterials based therapy. This is a >50% portion of the research and the authors have not addressed this. 

Reply 6:- in these review paper we have discussed about the Biotherapeutics approaches in targeting at biomolecular and cellular level in Diabetic Wound Healing. A new sub-section has been added which included biomaterials based therapy in treatment of diabetic wound healing under the section 9; subsection 9.5.

Reviewer 2 Report

This authors have described the mechanism of wound healing and how the diabetic wounds treatment differ from the normal wound. They talked about the process for wound healing: hemostasis, inflammatory stage, proliferative phase, remodeling and scar maturation, also all possible factors could affect the process. At the end they also claimed the bimolecular and cellular approaches for promoting diabetic wound healing. My overall impression is that the article includes a lot information in different sections, but hard to connect them. In the introduction, maybe more context about the whole article structure could be helpful. It described a lot about wound healing, but lack of contents about the difference between normal wound healing and diabetic healing. Specially for the description on the diabetic wound special system which could make it harder to treat. Also for the approaches in wound healing, sometimes this article talk both wound healing and diabetic wound healing, but sometimes only includes normal wound healing. That will be easier for readers to address the main point if authors can recontribute the contents for this two objects.

Some general comments/critiques I would like to make are following:

  1. In introduction, line 42, please reorganized the sentence: ‘which are hemostasis, inflammatory stage, proliferative phase, and remodeling, and scar maturation’
  2. In section 1.1, Fig 1 is showing too much information which is not included in the text. This figure shows two sections Wound healing and Tumor progress, but in the text there is only description about the mechanism, there is no explanation for the Tumor progress. Please simplify this figure if you only want to explain the mechanism or you can add more explanation for the full figure here
  3. In section 1.1, Fig 1, you may want to say ‘Hemostasis’, but it shows ‘Homeostasis’
  4. In section 1.2, line 10, please reorganize the sentence: ‘hemostasis, remodeling, replication, and inflammatory, step, which has a enduring…’
  5. In section 3, there should include more comparison for wound and diabetic wound at the beginning, it will be more clear when readers go to the subsections for this part
  6. Some acronyms (e.g. NGF, ROS) wither occur prior to their full form or never mentioned
  7. Some content in 8.4.2 seems like describing the function of miRNAs in wound healing, it should belong to section 6, please reorganize the two sections

Author Response

Reviewer 2

Comments and Suggestions for Authors

This authors have described the mechanism of wound healing and how the diabetic wounds treatment differ from the normal wound. They talked about the process for wound healing: hemostasis, inflammatory stage, proliferative phase, remodeling and scar maturation, also all possible factors could affect the process. At the end they also claimed the bimolecular and cellular approaches for promoting diabetic wound healing. My overall impression is that the article includes a lot information in different sections, but hard to connect them. In the introduction, maybe more context about the whole article structure could be helpful. It described a lot about wound healing, but lack of contents about the difference between normal wound healing and diabetic healing. Especially for the description on the diabetic wound special system which could make it harder to treat. Also for the approaches in wound healing, sometimes this article talk both wound healing and diabetic wound healing, but sometimes only includes normal wound healing. That will be easier for readers to address the main point if authors can recontribute the contents for this two objects.

Reply to Reviewer 2:-

  • Each section and subsections are interconnected to each other we have done some modification in our review paper which help them to interconnect with each other.
  • In earlier section of these review paper we have compare the normal wound healing with diabetic wound healing, how they are differ in occurrence and mechanism of healing process. That’s why we have talked about both normal wound healing and diabetic wound healing.

Some general comments/critiques I would like to make are following:

  1. In introduction, line 42, please reorganized the sentence: ‘which are hemostasis, inflammatory stage, proliferative phase, and remodeling, and scar maturation’

Reply 1:- We have rewrite the sentence and highlight it.

  1. In section 1.1, Fig 1 is showing too much information which is not included in the text. This figure shows two sections Wound healing and Tumor progress, but in the text there is only description about the mechanism, there is no explanation for the Tumor progress. Please simplify this figure if you only want to explain the mechanism or you can add more explanation for the full figure here

Reply 2:- We have modify our figure according to explanation given in the introduction.

  1. In section 1.1, Fig 1, you may want to say ‘Hemostasis’, but it shows ‘Homeostasis’

Reply 3:- Figure 1.1. Has been corrected and the word ‘Homeostasis’ is replaced with ‘Hemostasis’

  1. In section 1.2, line 10, please reorganize the sentence: ‘hemostasis, remodeling, replication, and inflammatory, step, which has an enduring…’

Reply 4:- Section 1.2, line 10 has been reorganize.

  1. In section 3, there should include more comparison for wound and diabetic wound at the beginning, it will be more clear when readers go to the subsections for this part

Reply 5:- a brief comparison for wound and diabetic wound has been included in the beginning of section 3.

  1. Some acronyms (e.g. NGF, ROS) wither occur prior to their full form or never mentioned

Reply 6:- Full form of the acronyms (e.g. NGF, ROS) has been mentioned in the first place where they occur.

  1. Some content in 8.4.2 seems like describing the function of miRNAs in wound healing, it should belong to section 6, please reorganize the two sections

Reply 7:- In section 6 we have separately describe the role of various miRNAs (progression and supression) in progression of the wound and their involvement in healing process.

Round 2

Reviewer 1 Report

The authors have made good effort to make positive changes in the manuscript in terms of content. 

However, there are still many English language errors that could benefit from editing. 

Some examples here:

1. The first sentence of the abstract is not grammatically correct. 

2. Line 33-34 is not scientifically or grammatically correct. 

3. Certain lines are written in present tense and certain in past tense: "It started when platelet and cytokine cause vasoconstriction and form a hematoma, reducing the loss of blood in the region affected."

4. Some letters are capitalized inconsistently and inaccurately. "Diabetic Wounds"

Author Response

Dear Reviewer

As per your suggestion  English language was improved.

Regards

Nagendra

Round 3

Reviewer 1 Report

There are still several errors in the English language. 

Author Response

Dear reviewer

As per your suggestion we will revised the english language.